# Atypical peripheral actin band formation via overactivation of RhoA and nonmuscle myosin II in mitofusin 2-deficient cells

Yueyang Wang[1], Lee D Troughton[2], Fan Xu[3,4], Aritra Chatterjee[3], Chang Ding[1], Han Zhao[5], Laura Pulido Cifuentes[1], Ryan B Wagner[6], Tianqi Wang[1], Shelly Tan[1], Jingjuan Chen[7], Linlin Li[3], David Umulis[3,8], Shihuan Kuang[7], Daniel M Suter[1,9,10], Chongli Yuan[5], Deva Chan[3], Fang Huang[3], Patrick W Oakes[2], Qing Deng[1,10,11]*

[1]Department of Biological Sciences, Purdue University West Lafayette, West Lafayette, United States; [2]Cell and Molecular Physiology, Loyola University Chicago, Chicago, United States; [3]Weldon School of Biomedical Engineering, Purdue University West Lafayette, West Lafayette, United States; [4]Advanced Research Institute of Multidisciplinary Science, Beijing Institute of Technology, Beijing, China; [5]Davidson School of Chemical Engineering, Purdue University West Lafayette, West Lafayette, United States; [6]School of Mechanical Engineering, Purdue University West Lafayette, West Lafayette, United States; [7]Department of Animal Sciences, Purdue University West Lafayette, West Lafayette, United States; [8]Department of Agricultural and Biological Engineering, Purdue University West Lafayette, West Lafayette, United States; [9]Purdue Institute for Integrative Neuroscience, Purdue University West Lafayette, West Lafayette, United States; [10]Purdue Institute for Inflammation, Immunology & Infectious Disease, Purdue University West Lafayette, West Lafayette, United States; [11]Purdue University Center for Cancer Research, Purdue University West Lafayette, West Lafayette, United States

*For correspondence:
deng67@purdue.edu

Competing interest: The authors declare that no competing interests exist.

**Abstract** Cell spreading and migration play central roles in many physiological and pathophysiological processes. We have previously shown that MFN2 regulates the migration of human neutrophil-like cells via suppressing Rac activation. Here, we show that in mouse embryonic fibroblasts, MFN2 suppresses RhoA activation and supports cell polarization. After initial spreading, the wild-type cells polarize and migrate, whereas the *Mfn2*[-/-] cells maintain a circular shape. Increased cytosolic $Ca^{2+}$ resulting from the loss of Mfn2 is directly responsible for this phenotype, which can be rescued by expressing an artificial tether to bring mitochondria and endoplasmic reticulum to close vicinity. Elevated cytosolic $Ca^{2+}$ activates $Ca^{2+}$/calmodulin-dependent protein kinase II, RhoA, and myosin light-chain kinase, causing an overactivation of nonmuscle myosin II, leading to a formation of a prominent F-actin ring at the cell periphery and increased cell contractility. The peripheral actin band alters cell physics and is dependent on substrate rigidity. Our results provide a novel molecular basis to understand how MFN2 regulates distinct signaling pathways in different cells and tissue environments, which is instrumental in understanding and treating MFN2-related diseases.

## Editor's evaluation

This important article presents evidence for a special role of mitofusin-2, not shared with mitofusin-1, in regulating the actin cytoskeleton through calcium, RhoA, and actomyosin contractility. The evidence is compelling and uses a variety of techniques, including creative approaches, to

investigate mitofusin-2's role in ER-mitochondrial tethering. The article will be of interest to investigators studying mitofusins, mitochondrial fission/fusion, and the actin cytoskeleton.

## Introduction

Cell spreading and migration play central roles in numerous physiological and pathophysiological processes. The dynamic cytoskeletal reorganization during cell migration is primarily achieved through a delicate balance between protrusive and retractive forces. The cytoskeleton and regulatory proteins cooperate with spatial and temporal precision to organize cell contents to control protrusions, adhesion, contractility, and force transmission (*Lauffenburger and Horwitz, 1996*; *Pollard and Borisy, 2003*; *Seetharaman and Etienne-Manneville, 2020*). The cytoskeletal networks are controlled by master regulators, such as the small Rho GTPases (*Nobes and Hall, 1995*; *Takai et al., 1995*; *Kaibuchi et al., 1999*). The initial cell spreading is driven by actin polymerization promoted by Rac1 and Cdc42 to form a sheet-like protrusion that generates a pushing force at the cell's leading edge. Subsequently, Ras homolog gene family member A (RhoA) and calcium/calmodulin (CaM)-dependent pathways modulate myosin-dependent contractile force by regulating focal adhesions (*Ridley and Hall, 1992*; *Nobes and Hall, 1999*; *Nobes and Hall, 1995*) and inducing the formation of actin-myosin filaments, which form stress fibers (*Ridley and Hall, 1992*; *Amano et al., 1996*). Reduced activity of RhoA is necessary for spreading and migration, which facilitates cell edge extension by reducing myosin-dependent contractile forces (*Wakatsuki et al., 2003*). Myosin, specifically nonmuscle myosin II (NMII), functions as a master regulator of cell stiffness, further influencing cell migration (*Tee et al., 2011*). Notably, in addition to generating mechanical force within a cell, NMII plays an essential role in sensing and responding to external forces applied to the cell (*Vicente-Manzanares et al., 2009*; *Aguilar-Cuenca et al., 2014*; *Lamb et al., 2021*).

Fibroblasts are mesenchyme-derived cells essential for tissue development and repair by remodeling the extracellular matrix. Additionally, they secrete multiple growth factors and respond to migratory cues such as PDGF (*Wynn, 2008*). As a widely used cell model, the in vitro motility of fibroblasts has been extensively studied. Local $Ca^{2+}$ pulses play critical roles in migrating cells, including fibroblasts, and $Ca^{2+}$ homeostasis controls the organization of the cytoskeleton spatially and temporally (*Bennett and Weeds, 1986*; *Tsai et al., 2015*; *Tsai and Meyer, 2012*). The intracellular $Ca^{2+}$ signals are predominantly generated from the intracellular $Ca^{2+}$ storage, the endoplasmic reticulum (ER), through inositol triphosphate (IP3) receptors (*Clapham, 2007*; *Parys and De Smedt, 2012*). Calmodulin (CaM) is an essential effector protein in cells to amplify the $Ca^{2+}$ signaling (*Clapham, 2007*). The $Ca^{2+}$/calmodulin (CaM)-dependent pathways promote the phosphorylation of the myosin light chain (MLC), promoting the formation of adhesive contacts and stress fibers (*Kamm and Stull, 1985*; *Stull et al., 1998*). In addition, $Ca^{2+}$/CaM activates $Ca^{2+}$/CaM kinases (CaMKs), including CaMKI, CaMKK, and CaMKII (*Saneyoshi and Hayashi, 2012*; *Soderling, 1999*; *Hudmon and Schulman, 2002*), each regulates actin cytoskeleton in distinct pathways (*Saneyoshi and Hayashi, 2012*). Notably, CaMKII bundles F-actin to remodel the cytoskeleton (*Lin and Redmond, 2008*; *Okamoto et al., 2007*; *O'Leary et al., 2006*) and regulates Rho GTPases, including Rac and RhoA, by phosphorylating their GEFs and GAPs (*Fleming et al., 1999*; *Okabe et al., 2003*; *Tolias et al., 2005*; *Xie et al., 2007*; *Penzes et al., 2008*).

Mitochondria are central cellular power stations. In addition, they regulate many physiological processes, such as maintaining intracellular $Ca^{2+}$ homeostasis and cell migration (*Denisenko et al., 2019*; *Campello et al., 2006*; *Zhao et al., 2013*; *Báthori et al., 2006*). The mitofusins (MFN1 and MFN2) localize to the outer mitochondrial membrane (OMM) and form homo- or heterodimers to promote mitochondrial outer membrane tethers (*Santel and Fuller, 2001*; *Chen et al., 2003*). Human MFN1 and MFN2 share ~80% similarity in protein sequence. They contain a large, cytosolic, N-terminal GTPase domain, two coiled-coil heptad-repeat (HR) domains, and two transmembrane domains (TM) crossing the OMM. MFN1 and MFN2 have primarily overlapping functions. Overexpression of either protein in MFN1 or MFN2 null cells promotes mitochondrial fusion (*Chen et al., 2003*). Knocking out either MFN1 or MFN2 leads to fragmented mitochondria in fibroblasts (*Chen et al., 2003*; *Cipolat et al., 2004*). Structural and biochemical studies revealed the difference between MFN1 and MFN2 in catalytic GTPase activity (*Ishihara et al., 2004*; *Efremov et al., 2019*) and in their ability to mediate trans-organelle calcium signaling (*Dorn, 2020*; *Naon et al., 2016*; *de Brito and Scorrano, 2008*).

MFN2, but not MFN1, localizes to the mitochondria-associated ER membranes (MAM) (*de Brito and Scorrano, 2008*; *Filadi et al., 2015*). Mfn2 ablation in various cell types increases the distance between the ER and mitochondria and severely reduces $Ca^{2+}$ transfer from the ER to mitochondria (*de Brito and Scorrano, 2008*; *Filadi et al., 2015*; *Naon et al., 2016*). Investigation of MFN2's role in human diseases has primarily focused on MFN-mediated mitochondrial fusion, trafficking, metabolism, mitophagy, and mitochondrial quality control. How MFNs regulate the cytoskeleton, however, remains unclear.

In our previous research to understand the importance of mitochondrial shape in neutrophil migration, we generated transgenic zebrafish lines with CRISPR-based neutrophil-specific knockout of mitochondrial fusion-related genes (*Maianski et al., 2002*; *Zhou et al., 2018*). Surprisingly, we noticed a phenotype specific to Mfn2 deletion: most neutrophils exited the hematopoietic tissue and circulated in the bloodstream in homeostasis. We further demonstrated that MFN2 regulates neutrophil adhesive migration and Rac activation using the human neutrophil-like differentiated HL-60 cells (*Zhou et al., 2020*). Although we identified an essential role for MFN2 in neutrophil adhesion and migration, it is unclear how MFN2 regulates actin cytoskeleton organization and other cellular behaviors, such as cell spreading.

Here, we used mice embryonic fibroblasts (MEFs) as a model to further characterize how MFN2 regulates cytoskeletal organization. We demonstrate that MFN2 regulates cytoskeletal organization by suppressing Rho and NMII activity. *Mfn2* depletion upregulates cytosolic $Ca^{2+}$ in MEFs, leading to RhoA and NMII overactivation and forming a prominent 'peripheral actin band (PAB)' structure. This PAB hampered cell adhesive migration and caused significant changes in mechanical properties, including cell stiffness and membrane tension. Together, our results provided an in-depth molecular understanding of the role of MFN2 in cytoskeleton dynamics, cell spreading, and adhesive migration, which may lead to a better understanding and treatment of MFN2-associated diseases.

## Results

### MFN2 deficiency changes cell morphology and impairs adhesive 2D random migration in MEFs

As a first step in investigating the role of MFN2 in MEF cells, we confirmed the respective protein loss in indicated cell lines by immunoblotting (*Figure 1A*). We first analyzed the morphology and spread area of the cells in the culture. We found that the average cell spread area was reduced significantly in *Mfn2*-null MEFs (1303 µm²) compared to *wt* (2233 µm²) and *Mfn1*-null MEFs (2350 µm²) (*Figure 1B*). *Mfn2*-null MEFs also displayed significantly increased cell circularity (*Figure 1C*). To evaluate the function of MFN2 protein in the cytoskeleton and cell migration, we seeded the cells on chamber slides. We imaged them overnight using phase-contrast, time-lapse microscopy. In *Mfn2*-null MEFs, cell motility (0.23 ± 0.08 µm/min) was significantly reduced compared to *wt* (0.53 ± 0.16 µm/min) or *Mfn1*-null MEFs (0.49 ± 0.12 µm/min) (*Figure 1D–F*, *Video 1*). No significant change in directionality was observed in *Mfn2*-null MEFs (*Figure 1G*). During cell spreading, *wt* and *Mfn1*-null MEFs generated rapid protrusive filopodia and lamellipodia, eventually elongating to form traditional fibroblast-like shapes and began to migrate. However, the elimination of MFN2 caused significant defects in elongation, and the cells remained rounded (*Figure 1H and I*). The morphological differences became apparent during the spreading process, especially after 20 mins. *Mfn2*-null MEFs only extended round membrane ruffles but did not simultaneously form multiple short lamellae separated by concave edges. Immunofluorescence also revealed striking differences in actin stress fiber organization. Both *wt* and *Mfn1*-null MEFs displayed parallel stress fibers in the cell body, while *Mfn2*-null MEFs contained an enrichment in actin filaments in the cell cortex with reduced stress fibers at the center of the cells (*Figure 1—figure supplement 1A*). To rule out the possibility of side effects caused by long-term culture, we isolated MEFs from *Mfn2^{flox/flox}* mice. The addition of Cre-expressing adenovirus induced loss of MFN2 within 48 hr, reproduced the rounded morphology, and altered actin cytoskeleton organization seen in *Mfn2*-null MEFs (*Figure 2—figure supplement 1B–D*).

To further confirm the functional role of the MFN2 on cytoskeletal organization and cell migration, we re-expressed MFN1 or MFN2 in *Mfn2*-null MEFs (*Figure 2A*). Only MFN2 re-expression significantly increased cell motility (0.32 ± 0.18 µm/min), comparing to *Mfn2*-null MEFs (0.22 ± 0.18 µm/min), or those with MFN1 re-expression (0.21 ± 0.15 µm/min) (*Figure 2B–D*, *Video 2*). Notably, MFN2

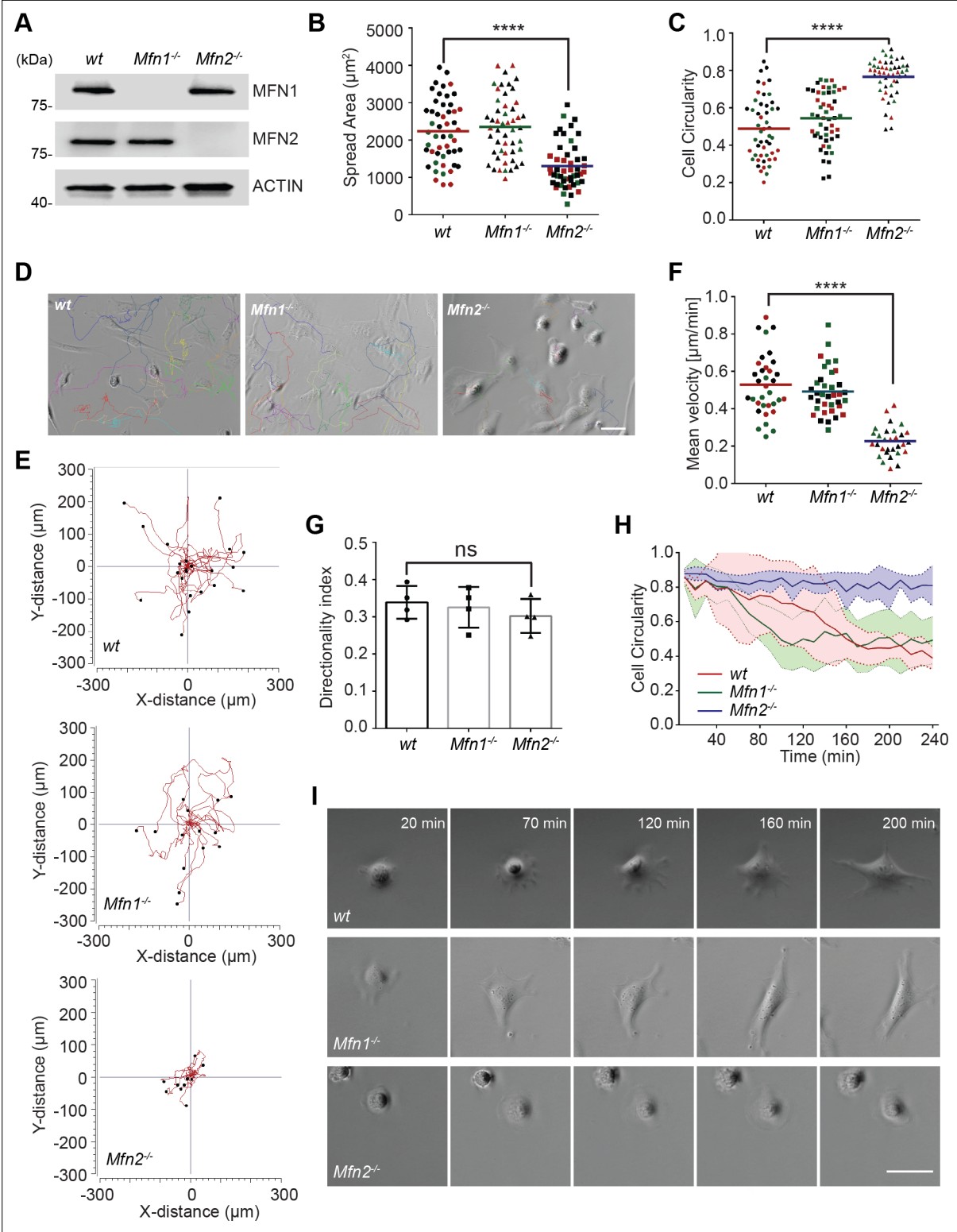

**Figure 1.** MFN2 regulates random migration and spreading in mice embryonic fibroblasts (MEFs). (**A**) Western blot determining the expression levels of MFN1 and MFN2 in wt, Mfn2-null, and Mfn1-null MEFs. (**B, C**) Spread area (**B**) and circularity (**C**) of wt, Mfn1-null, and Mfn2-null MEFs after overnight culture. The individual points represent individual MEF cells. (**D–G**) representative images with individual tracks (**D**), Wind–Rose plots (**E**), quantification of velocity (**F**), and directionality (**G**) of wt, Mfn1-null, and Mfn2-null MEFs cells during random migration. (**H, I**) Quantification of cell circularity (**H**) and representative images (**I**) of indicated MEFs during cell spreading at indicated time points. Data are presented as mean ± SD in (**F**) and were pooled from a total of 18 cells in three independent experiments. Bars represent arithmetic means ± SD. One representative result of three biological repeats is

*Figure 1 continued on next page*

*Figure 1 continued*

shown in (**A, D, E, I**). Data are pooled from three independent experiments in (**B, C, F, G**). n = 50 cells are tracked and counted in (**B, C**). N = 30 cells are quantified in (**D**). ****p<0.0001 (one-way ANOVA). Scale bars: 50 μm.

The online version of this article includes the following source data and figure supplement(s) for figure 1:

**Source data 1.** Original blots and figures with the bands labeled for *Figure 1A*.

**Figure supplement 1.** MFN2 deficiency changes mice embryonic fibroblast (MEF) morphology.

re-expression also rescued the cell's ability to polarize during the spreading process (*Figure 2E*), increased cell area, and decreased circularity (*Figure 2F and G*). Similarly, doxycycline (DOX)-induced re-expression of MFN2 for 48 hr in *Mfn2*-null MEFs also significantly restored the actin cytoskeleton organization and cell morphology (*Figure 2J*, *Figure 3—figure supplement 1E and F*). Additionally, re-expressing MFN2, but not MFN1, in *Mfn2*-null MEFs restored the tubular mitochondrial network (*Figure 1- figure supplements 1G*). Together, these results suggest that these cells' morphological reorganizations and migratory defects are specifically caused by the loss of MFN2 protein in MEFs.

The significant differences in actin architecture in *Mfn2*-null MEFs, an atypical PAB, could account for the spreading and migratory defects (*Figure 5—figure supplement 1A and B* and *Figure 2H*). We used the ImageJ plugin FiloQuant (*Jacquemet et al., 2019*) to quantify the cells with PAB structure. We developed an algorithm to calculate the percentage of actin in the cell border region (*Figure 2— figure supplement 1A and B*). If the cell's border region contains more than 50% actin, and the cellular circularity is higher than 0.6, we consider it a 'PAB' cell. *Mfn2*-null MEFs displayed a significantly higher 'PAB' rate (33.2%), while MFN2 re-expression reduced the average 'PAB' rate to 16.3% (*Figure 2H and I*).

## Increased cytosolic Ca²⁺ suppresses cell migration in *Mfn2*-null MEFs

These striking alternations in cell morphology, spreading, and migration prompted us to investigate the downstream effectors of MFN2. We focused on one of the MFN2-specific functions: maintaining cellular Ca²⁺ homeostasis by tethering ER and mitochondria. It was previously reported that the increased distance between ER and mitochondria in the absence of MFN2 elevates cytosolic Ca²⁺ transients in MEFs (*de Brito and Scorrano, 2008*; *Filadi et al., 2015*; *Naon et al., 2016*). We, therefore, measured cytosol Ca²⁺ in response to PDGF-BB stimulation and confirmed the previous observation (*Figure 3A*). To evaluate the effect of cytosolic Ca²⁺ accumulation on MEF cell migration, we treated *wt* MEFs with the calcium ionophore A23187, an ion carrier facilitating Ca²⁺ transport across the plasma membrane. A23187 increased cytosol Ca²⁺ levels in *wt* MEFs (*Figure 3—figure supplement 1A*). A reduction in cell migration speed was observed in a dose-dependent manner, which phenocopied the motility reduction in *Mfn2*-null MEFs (*Figure 3B*). The treatment of A23187 increased the percentage of actin in the cell border region. However, it failed to phenocopy the 'PAB' structure in *wt* MEFs (*Figure 3—figure supplement 1C–E*), indicating that excessive cytosolic Ca²⁺ is not sufficient to induce the 'PAB' structure.

On the other hand, the 'PAB' structure was rescued by inhibiting cytosolic Ca²⁺ in *Mfn2*-null MEFs with an intracellular calcium chelator BAPTA-AM (*Figure 3—figure supplement 1B–E*, *Figure 3C and D*, *Video 3*). Consistently, the reduced motility in *Mfn2*-null MEFs was partially rescued, suggesting that the PAB structure and defects on migration described above in *Mfn2*-null MEFs can be explained, at least in part, by excessive cytosolic Ca²⁺.

To confirm that MEF2 regulates cytosolic Ca²⁺ via maintaining ER-mitochondria tether, we introduced an artificial tether construct (*Kornmann et al., 2009*) into the *Mfn2*-null MEFs (*Mfn2⁻/⁻*+T). The tether comprises a GFP protein carrying ER and mitochondrial localization sequences at opposite ends, which functions independently of MFN2. We adopted an established probe to detect the ER-mitochondria contacts (*Vallese*

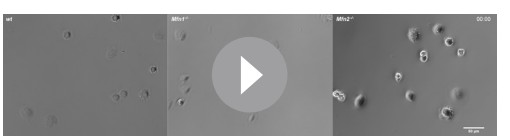

**Video 1.** MFN2 regulates the migration and spreading of mice embryonic fibroblasts (MEFs). Cell spreading and random migration of *wt*, *Mfn2*-null, and *Mfn1*-null MEFs in the μ-slide 15 min after plating. Time-lapse images were taken every 10 min for 16 hr and 40 min. Individual MEFs were tracked for velocity quantification. Scale bar: 50 m.

https://elifesciences.org/articles/88828/figures#video1

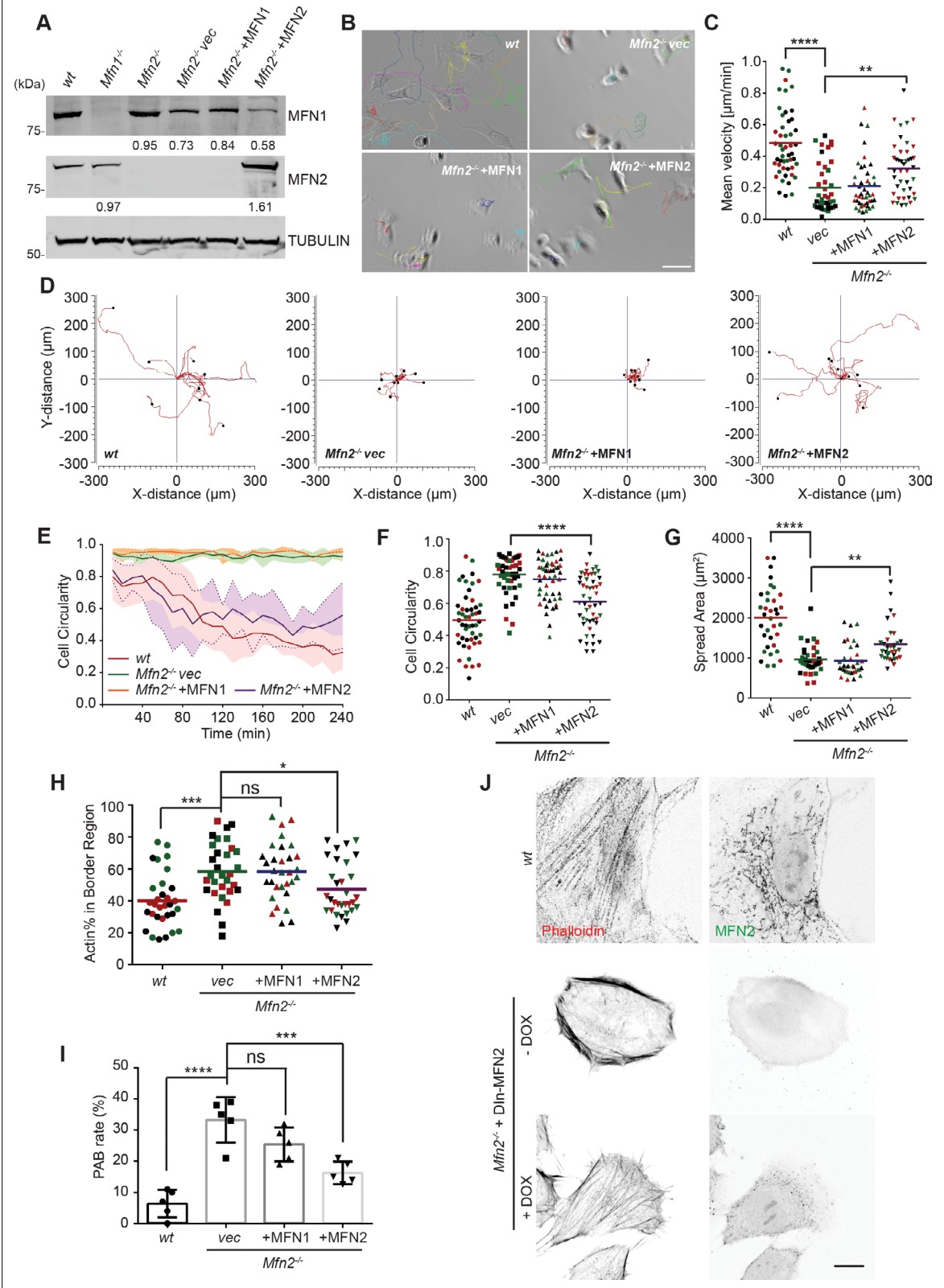

**Figure 2.** Cre expression of MFN2 rescues random migration and spreading defects in Mfn2-null mice embryonic fibroblasts (MEFs). (**A**) Western blot determining the expression level of MFN1 and MFN2 in indicated MEF cells. Percentages of knockdown or re-expression were calculated by normalizing the intensity to vinculin first, then normalizing to the wt group. (**B–D**) Representative images with individual tracks (**B**), quantification of velocity (**C**), and Wind–Rose plots (**D**) of indicated MEF cells during random migration. (**E**) Quantification of cell circularity of wt and Mfn2-null MEFs

*Figure 2 continued on next page*

*Figure 2 continued*

with vec, MFN1, or MFN2 re-expressed during spreading at indicated time points. Data are presented as mean ± SD in (**E**) (n = 5). (**F, G**) Cell circularity (**F**) and cell spreading area (**G**) of indicated MEFs measured after overnight culture. (**H, I**) Percentage of Actin abundance in the cell border region (**H**) and peripheral actin band (PAB) cell percentage in each view was quantified using our custom algorism (see *Figure 2-figure supplement 1*). (**J**) Representative images of wt, Mfn2-null with doxycycline-induced MEF2 (DIn-MFN2) MEF cells treated with or without doxycycline for 48 hr. The cells are immunostained with phalloidin and MFN2. One representative result of three biological repeats is shown in (**A, B, D, H**). Data are pooled from three independent experiments in (**C, F–I**). n = 30 cells are tracked and counted in (**C**); n = 35 cells are quantified in (**F–H**). Five different views from three biological repeats are quantified in (**I**). *p≤0.05, **p≤0.01, ***p≤0.001, ****p<0.0001 (one-way ANOVA in **C, E, F**, unpaired t-test in **H, I**). Scale bars: 50 μm in (**B**), 10 μm in (**J**).

The online version of this article includes the following source data and figure supplement(s) for figure 2:

**Source data 1.** Original blots and figures with the bands labeled for *Figure 2A*.

**Figure supplement 1.** Custom algorithm to identify cells with peripheral actin band (PAB).

et al., 2020). In line with previous studies, *Mfn2*-null MEFs displayed significantly decreased ER-mitochondria contacts and MFN1 re-expression failed to restore the phenotype. Expressing MFN2 or the artificial tether construct restored ER-mitochondria contacts in *Mfn2*-null MEFs (*Figure 3—figure supplement 2A and B*). Indeed, expressing the tether corrected the cytosolic $Ca^{2+}$ levels in response to PDGF-BB stimulation partially rescued the migration speed in *Mfn2*-null MEFs (*Figure 3F–H*, *Video 4*) and decreased PAB cell percentage (*Figure 3—figure supplement 2C*). These data suggest that MFN2 regulates cell morphology and adhesive migration in MEFs by maintaining mitochondria-ER interaction and $Ca^{2+}$ homeostasis.

## Elevated CaMKII activation is associated with MFN2-regulated random migration

Given that cytosolic $Ca^{2+}$ plays essential roles in MFN2-mediated cytoskeleton regulation and cell migration, we looked at the kinases and phosphatases regulated by $Ca^{2+}/$ calmodulin (CaM), including CaMKK, CaMKII, and calcineurin, which are previously shown to regulate actin bundling (*Saneyoshi and Hayashi, 2012*; *Figure 4A*). AIP, the CaMKII inhibitor, partially restored motility in *Mfn2*-null MEFs. Neither the calcineurin inhibitor FK506 nor the CaMKK inhibitor STO609 had any effect (*Figure 4B and C*, *Video 5*). We found a higher level of phosphorylated or active CaMKII in *Mfn2*-null MEFs, which was also reduced by the expression of MFN2 or the tether (*Figure 4D and E*). The function of CamKII was further confirmed by expressing a wild-type (CaMKII-WT) or a dominant negative (CaMKII-DN) version of CamKII in *Mfn2*-null MEFs. CaMKII-DN, but not CaMKII-WT, induced a moderate but significant increase in migration velocity (*Figure 4F and G*, *Video 6*). Together, our results indicate a role of CaMKII in cytoskeleton regulation in *Mfn2*-null MEFs.

## MFN2 deficiency-induced migration defect is independent of Rac and CDC42

The Rho GTPase family members are master regulators of the actin cytoskeleton and cell migration. The peripheral enrichment of actin filaments and extensive membrane ruffles we observed in *Mfn2*-null MEF cells resembled the classic phenotype seen in fibroblasts with constitutively active Rac (*Hall, 1998*). Therefore, we hypothesized that Rac might be overactivated in MFN2-depleted MEF cells. We performed a RAC-GTP pulldown and observed no significant increase in the absence of MFN2 (*Figure 5—figure supplement 1A and B*). In addition, the Rac inhibitor, CAS1090893, did not rescue the cell motility defect of the *Mfn2*-null

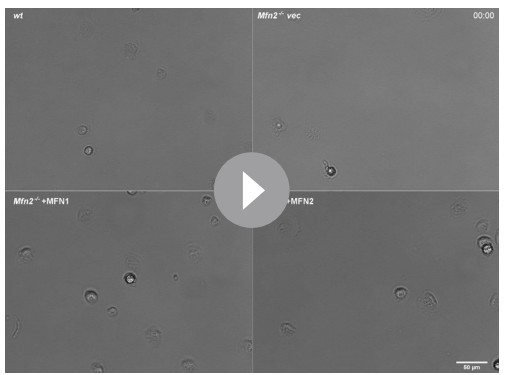

**Video 2.** MFN2 re-expression, but not MFN1, restores migratory defects in *Mfn2*-null mice embryonic fibroblasts (MEFs). Cell spreading and random migration of wt and *Mfn2*-null MEFs with vec, MFN1, or MFN2 re-expressed in the μ-slide. Time-lapse images were taken every 10 min for 14 hr and 50 min. Individual MEFs were tracked for velocity quantification. Scale bar: 50 m.

https://elifesciences.org/articles/88828/figures#video2

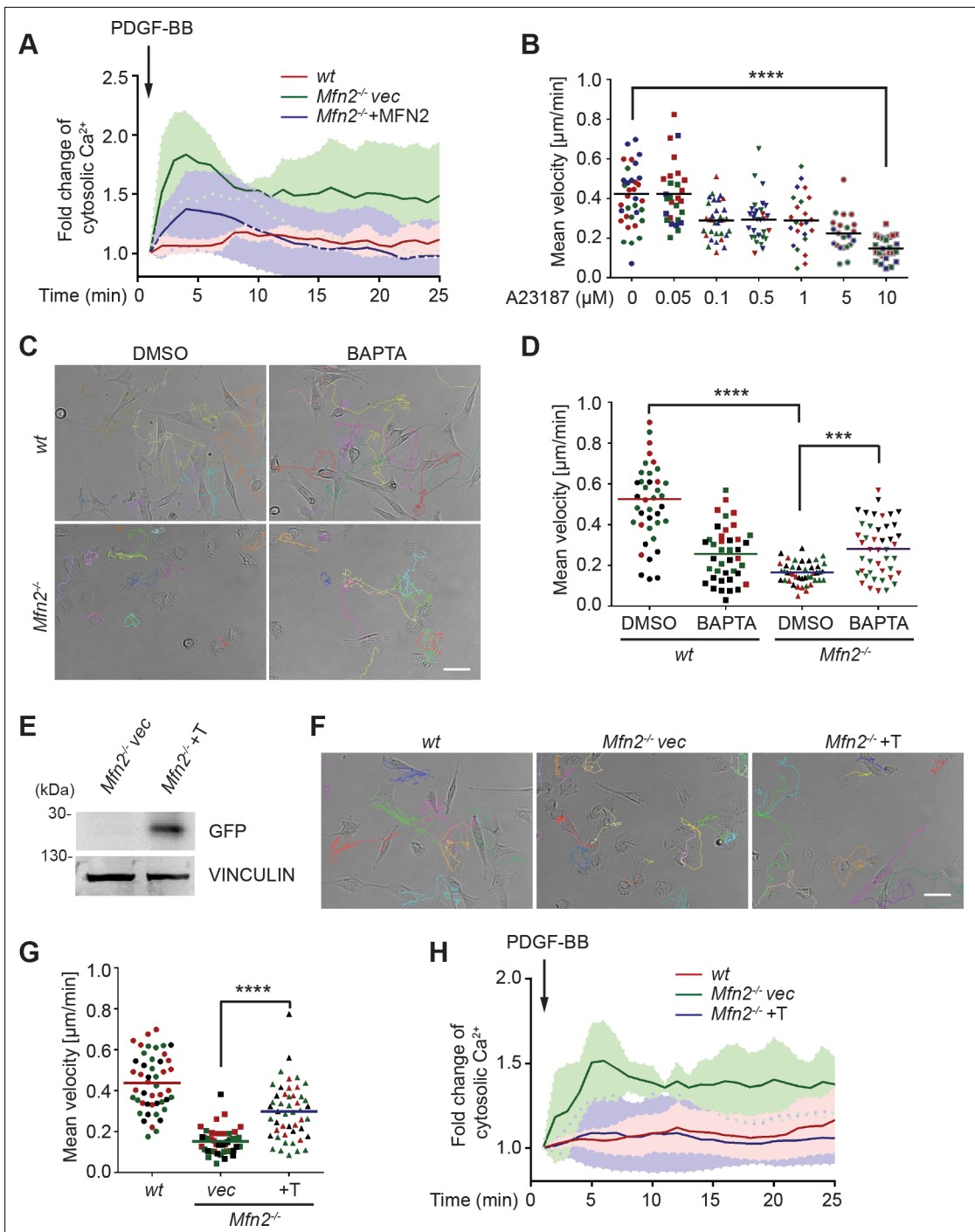

**Figure 3.** CMFN2 regulates random migration through cytosolic Ca²⁺ and endoplasmic reticulum (ER)-mitochondria tether. (**A**) Fluo-4 recordings of cytosolic Ca²⁺ in the indicated cell lines after PDGF-BB stimulation. (**B**) Quantification of the velocity of *wt* mice embryonic fibroblast (MEF) random migration in the presence of vehicle or different concentrations of the Ca²⁺ ionophore A23187. (**C, D**) Representative images with individual tracks (**C**) and quantification of velocity (**D**) of wt or Mfn2-null MEFs during random migration with or without the presence of the intracellular calcium chelator BAPTA-AM. (**E**) Western blot of GFP in indicated cell lines. Mfn2⁻/⁻+T, Mfn2-null MEFs with synthetic ER-GFP-mitochondria tether construct. (**F, G**) Representative images with individual tracks (**F**) and quantification of velocity (**G**) of indicated MEF cells. (**H**) Fluo-4 recordings of cytosolic Ca²⁺ in the indicated cell lines after PDGF-BB stimulation. The individual points in (**B, D, G**) are the mean speeds for individual MEF cells. Data are presented as mean ± SD in (**A, H**). Data are pooled from three independent experiments in (**A, H**). One representative result of three biological repeats is shown in (**C–G**). n = 30 cells are tracked and counted in (**B, G**). n = 35 cells are tracked and measured in (**D**). ***p<0.001, ****p<0.0001 (one-way ANOVA in **B, G** and two-way ANOVA in **D**). Scale bars: 50 µm.

*Figure 3 continued on next page*

*Figure 3 continued*

The online version of this article includes the following source data and figure supplement(s) for figure 3:

**Source data 1.** Original blots and figures with the bands labeled for *Figure 3E*.

**Figure supplement 1.** Excessive cytosolic Ca$^{2+}$ is insufficient to phenocopy the peripheral actin band ('PAB') structure in *wt* mice embryonic fibroblasts (MEFs), while the cytosolic Ca$^{2+}$ inhibitor BAPTA rescues 'PAB' structure in *Mfn2*-null MEFs.

**Figure supplement 2.** Expression of MEF2 or an endoplasmic reticulum (ER)-mitochondria tether construct restores ER-mitochondria contacts and mitochondria morphology.

MEFs (*Figure 5-figure supplement 1C*), suggesting that Rac is not the primary effector regulated by MFN2. CDC42-GTP levels are comparable between the *wt* and *Mfn2*-null cells (*Figure 5—figure supplement 1D and E*).

## Loss of MFN2 drives overactivation of RhoA GTPase and redistribution of focal adhesions

In contrast to the minor changes in Rac and CDC42 activity, we detected a marked increase of RhoA-GTP in *Mfn2*-null MEFs (*Figure 5A and B*). Restoring MFN2 expression or introducing the ER-mitochondria artificial tether brought the RhoA activation level back to the *wt* level. We then compared the distribution of focal adhesion protein paxillin (Pax) by immunofluorescence. The focal adhesions were fewer and restricted to the cell periphery in *Mfn2*-null MEFs (*Figure 5C and D*). In addition to the striking PAB architecture described previously, the focal adhesion complexes in *Mfn2*-null MEFs appeared larger, consistent with the observation of RhoA overactivation in fibroblasts (*Hall, 1998*; *Figure 5E*). We plated the cells on fibronectin, fibrinogen, or uncoated cover glasses. We discovered that the PAB formation is independent of the extracellular substrates (*Figure 6—figure supplement 1A*). These focal adhesion differences hinted that the spreading and migration defects in cells depleted of MFN2 might be explained, at least in part, by significantly increased RhoA activity. Moreover, BAPTA-AM treatment or DOX-induced MFN2-re-expression reduced the heightened activity of RhoA in *Mfn2*-null MEFs (*Figure 5F and G*), indicating that the cytosolic Ca$^{2+}$ increase was responsible for the RhoA overactivation in *Mfn2*-null MEFs.

## Increased myosin regulatory light chain activity is critical for PAB formation in *Mfn2*-null MEFs

To fully understand the downstream mechanism of MFN2 in regulating the cytoskeleton, we used pharmacological inhibitors for proteins regulated by CaM (*Figure 6A*, *Figure 6—figure supplement 1B and C*, *Video 7*). The RhoA inhibitor I, the Rho-associated protein kinase (ROCK) inhibitor Y27632, the myosin inhibitor Blebbistatin, and the myosin light chain kinase (MLCK) inhibitor ML-7 showed the most pronounced effects on restoring both the motility and cell morphology in *Mfn2*-null

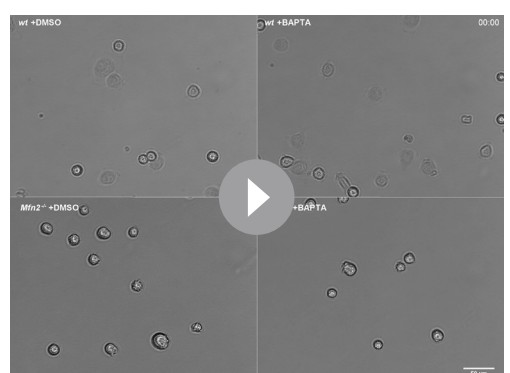

**Video 3.** Cytosolic Ca$^{2+}$ inhibition restores the migration defects in *Mfn2*-null mice embryonic fibroblast (MEF) cells. Cell spreading and random migration of *wt* and *Mfn2*-null MEFs treated with DMSO or BAPTA-AM (20 μM) in the μ-slide. Time-lapse images were taken every 10 min for 14 hr and 30 min. MEFs were tracked for velocity quantification. Scale bar: 50 μm.

https://elifesciences.org/articles/88828/figures#video3

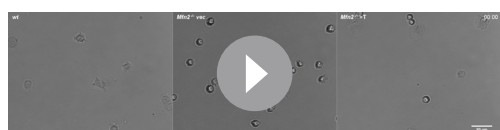

**Video 4.** Restoring the endoplasmic reticulum (ER)-mitochondrial tether rescues the migration defects in *Mfn2*-null mice embryonic fibroblast (MEF) cells. Cell spreading and random migration of *wt*, *Mfn2*-null MEFs with *vec* or synthetic tether construct in the μ-slide. Time-lapse images were taken every 10 min for 17 hr and 50 min. Individual MEFs were tracked for velocity quantification. Scale bar: 50 m.

https://elifesciences.org/articles/88828/figures#video4

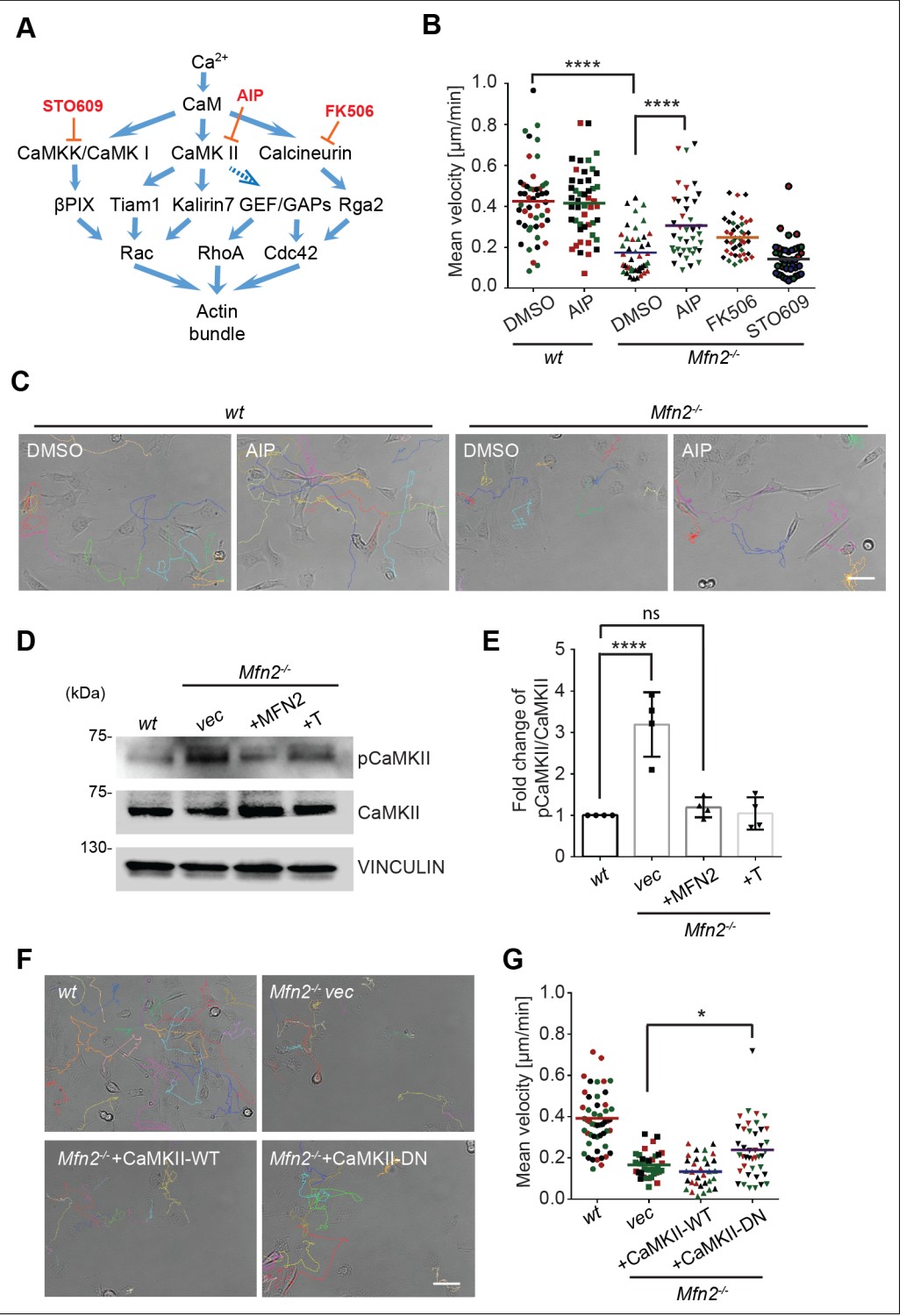

**Figure 4.** CaMKII activation mediates Mfn2 deficiency-induced inhibition in mice embryonic fibroblast (MEF) migration. (**A**) Selected signaling cascades involved in the regulation of the actin cytoskeleton via $Ca^{2+}$. Blue arrows indicate positive regulation. Dashed blue arrows indicate positive regulation with unclear mechanisms. Orange T-shaped bars indicate negative regulation of the pharmacological inhibitors. (**B**) Quantification of the velocity of indicated MEF cells treated with DMSO, the CaMKII inhibitor AIP, the Calcineurin inhibitor FK506, or the CaMKK inhibitor STO609. (**C**) Representative images with individual tracks of wt or Mfn2-null MEFs treated with the CaMKII inhibitor AIP. (**D, E**) Western blot (**E**) and quantification (**F**) determining the amount of pCaMKII and pan-CaMKII in wt, Mfn2-null MEFs with vec, CaMKII-WT, or CaMKII-DN overexpressed after treating with 25 µM PDGF-BB for

*Figure 4 continued on next page*

*Figure 4 continued*

4 min. (**F, G**) Quantification of velocity (**G**) and representative images with individual tracks (**F**) of wt, Mfn2-null MEFs with vec, CaMKII-WT, or CaMKII-DN overexpressed during random migration. One representative result of three biological repeats is shown in (**B, C, F, G**). Four biological repeats are shown in (**E**). n = 40 cells are quantified in (**B, G**). *p≤0.05, ***p≤0.001, ****p<0.0001 (one-way ANOVA in **E, G** and two-way ANOVA in **B**). Scale bars: 50 μm.

The online version of this article includes the following source data for figure 4:

**Source data 1.** Original blots and figures with the bands labeled for *Figure 4D*.

MEFs (*Figure 6A and B*, *Figure 6—figure supplement 1B and C*, *Video 5*). The focal adhesion kinase inhibitor 14 (FAK inhibitor 14), the LIM kinase (LIMK) inhibitor BMS-5, and the Arp2/3 inhibitor CK-666 had no statistically significant effects on restoring cell motility (*Figure 6—figure supplement 1B*). This result was confirmed by analyzing the actin filament organization and focal adhesion distribution in inhibitor-treated *Mfn2*-null MEFs. RhoA inhibitor-I, ML-7, Y27632, or Blebbistatin abrogated the PAB in *Mfn2*-null MEFs and restored typical fibroblast characteristics, including the formation of filopodia, developed cell edges, and focal adhesions at the leading and trailing edges (*Figure 6C*). Taken together, these findings demonstrate a potential mechanism whereby excessive cytosolic Ca$^{2+}$ in the absence of MFN2 leads to the overactivation of RhoA and MLCK, which increases MLC activity and contributes to the PAB in *Mfn2*-null MEFs (*Figure 6D*, *Figure 6—figure supplement 1D*).

Given the essential role of the MLC, we probed for pMLCII in *wt*, *Mfn1*-null, and *Mfn2*-null MEFs. As expected, the pMLCII level was significantly increased in *Mfn2*-null MEFs (*Figure 7A and B*), which could be corrected by expressing MFN2 or the artificial tether in *Mfn2*-null MEFs (*Figure 7C and D*). Using immunofluorescence microscopy, we noted that pMLCII colocalized with peripheral actin bundles in *Mfn2*-null MEFs (*Figure 7E*). To confirm the importance of MLC activity, we knocked down the two MLC kinases, MLCK or ROCK, in *Mfn2*-null MEFs (*Figure 7F*). Both knockdown cell lines displayed significantly reduced pMLCII levels (*Figure 7G*), restored stress fiber architecture, and redistributed focal adhesions (*Figure 7H*). Both knockdown lines also showed significantly decreased PAB cell percentages with increases in cell spread area and more polarized morphology (*Figure 7I–L*).

As MFN2 is a mitochondrial protein, we also performed a seahorse assay to measure mitochondrial functions in each cell line. *Mfn2*-null MEFs showed a decreased rate of oxidative phosphorylation relative to *wt* and *Mfn1*-null MEFs (*Figure 7—figure supplement 1A and B*), while MFN2 re-expression or introducing ER-mitochondrial tether increased the overall oxygen consumption rate in *Mfn2*-null MEFs (*Figure 7—figure supplement 1C and D*). Knocking down MLCK in *Mfn2*-null MEFs enhanced oxidative metabolism, whereas ROCK knockdown reduced oxygen consumption (*Figure 7—figure*

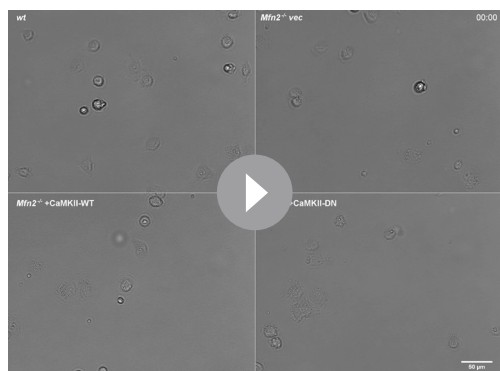

**Video 5.** CaMKII inhibition rescues the migration defects in *Mfn2*-null mice embryonic fibroblast (MEF) cells. Cell spreading and random migration of *wt* and *Mfn2*-null MEFs treated with DMSO or AIP (40 μM) in the μ-slide. Time-lapse images were taken every 10 min for 17 hr and 50 min. MEFs were tracked for velocity quantification. Scale bar: 50 μm.

https://elifesciences.org/articles/88828/figures#video5

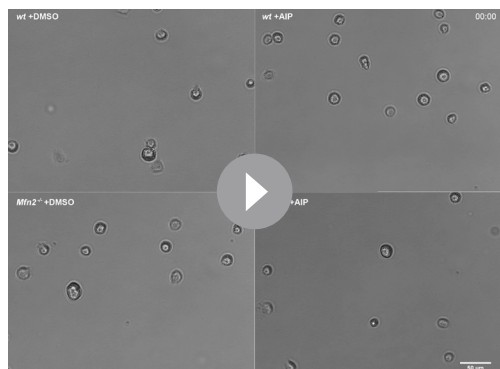

**Video 6.** Expression of CaMKII-DN rescues the migration defects in *Mfn2*-null mice embryonic fibroblast (MEF) cells. Cell spreading and random migration of *wt*, *Mfn2*-null MEFs with *vec*, CaMKII-WT, or CaMKII-DN in the μ-slide. Time-lapse images were taken every 10 min for 17 hr and 50 min. MEFs were tracked for velocity quantification. Scale bar: 50 μm.

https://elifesciences.org/articles/88828/figures#video6

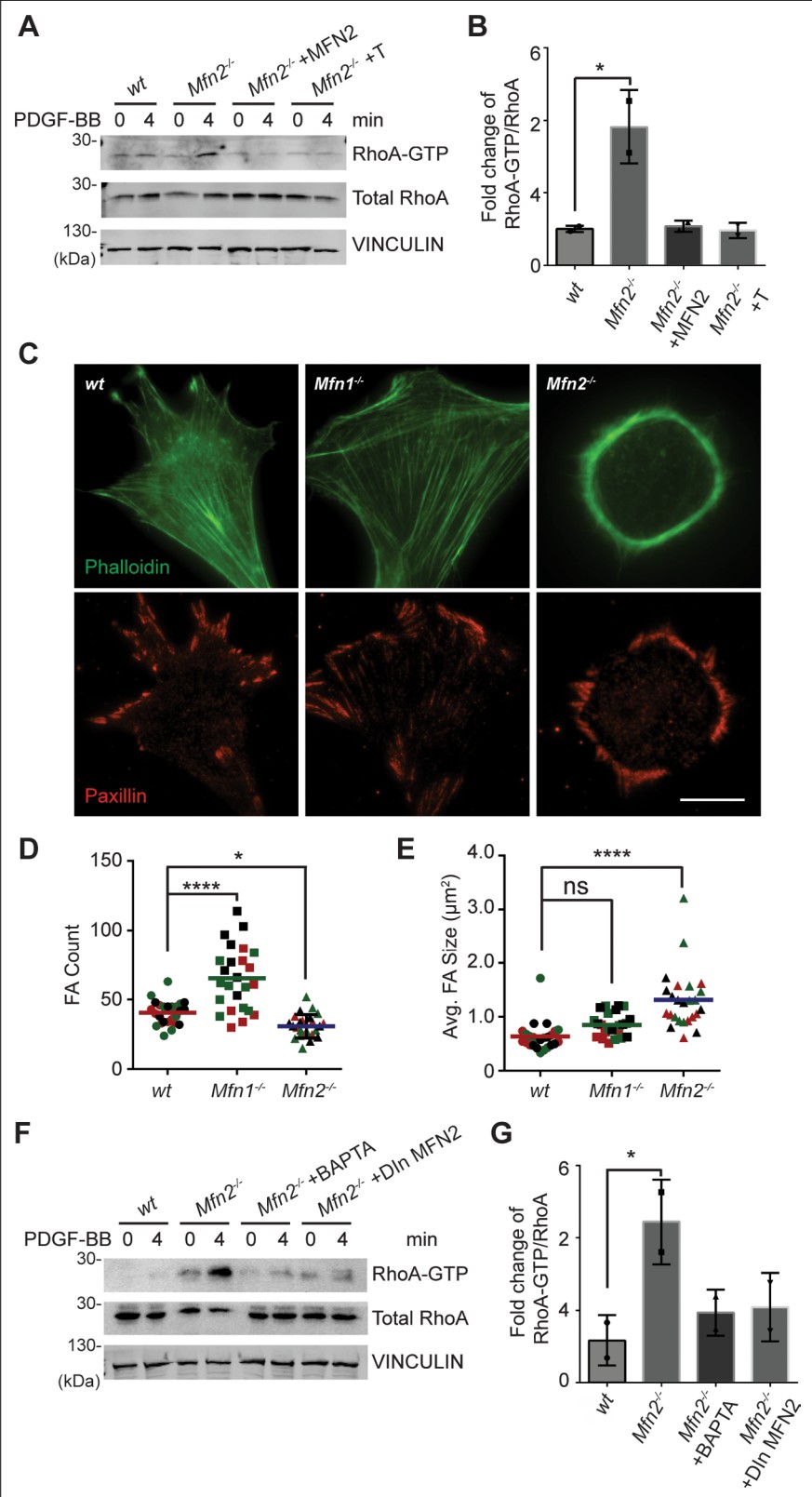

**Figure 5.** Loss of MFN2 induces heightened RhoA activation in mice embryonic fibroblasts (MEFs). (**A, B**) RhoA pulldown activation assay demonstrates increased RhoA-GTP in *Mfn2*-null MEFs, which can be corrected by re-expressing MFN2 or inducing a mitochondria-endoplasmic reticulum (ER) tether. (**A**) Western blot and (**B**) quantification determining the amount of RhoA-GTP and total RhoA protein in *wt*, *Mfn2*-null MEFs, *Mfn2*-null

*Figure 5 continued on next page*

*Figure 5 continued*

MEFs with MFN2 re-expression, or with an artificial ER-mitochondria tether; the indicated cell lines were treated with 25 ng/ml PDGF-BB for 0 or 4 min. (**C**) Immunofluorescence of F-actin (phalloidin) and Paxillin in *wt*, *Mfn1*-null, and *Mfn2*-null MEFs after overnight culture. (**D, E**) *Mfn2*-null MEFs display slightly decreased FA numbers (**D**) but significantly larger FA sizes (**E**). (**F, G**) RhoA pulldown activation assay demonstrates cytosolic $Ca^{2+}$ inhibition corrects RhoA-GTP level in *Mfn2*-null MEFs. (**F**) Western blot and (**G**) quantification showing the amount of RhoA-GTP protein in *wt*, *Mfn2*-null MEFs, *Mfn2*-null MEFs treated with BAPTA, or *Mfn2*-null MEFs with doxycycline (DOX)-induced MFN2 re-expression for 48 hr; the indicated cells were treated with 25 ng/ml PDGF-BB for the indicated time. RhoA-GTP/total RhoA ratios at 4 min were normalized to 0 min to show the fold changes in (**B, G**). n = 30 cells were quantified in (**D, E**). One representative result of two biological repeats is shown in (**A, B, F, G**). *$p \leq 0.05$ (one-way ANOVA comparing each group with the average of the *wt* group). Scale bar: 50 μm.

The online version of this article includes the following source data and figure supplement(s) for figure 5:

**Source data 1.** Original blots and figures with the bands labeled for *Figure 5A*.

**Source data 2.** Original blots and figures with the bands labeled for *Figure 5F*.

**Figure supplement 1.** Activities of Rac and Cdc42 are not increased in MFN2-deficient mice embryonic fibroblasts (MEFs).

**Figure supplement 1—source data 1.** Original blots and figures with the bands labeled for *Figure 5—figure supplement 1A*.

**Figure supplement 1—source data 2.** Original blots and figures with the bands labeled for *Figure 5—figure supplement 1D*.

---

supplement 1E and F). Given that MLCK and ROCK knockdown can restore cell spreading and migration in *Mfn2*-null MEFs, alterations in mitochondrial metabolism are unlikely to be the primary determinant in PAB formation.

## Myosin regulatory light chain overexpression phenocopied MFN2 depletion

Mitochondria and MFN2 regulate multiple cellular signaling pathways in addition to cytosolic calcium. To determine whether the PAB is primarily driven by RhoA or MLCK activation, we attempted to constitute the PAB in *wt* MEFs. We first tried Rho activator treatment or expression of constitutively active MLCK (MLCK-CA) (*Wong et al., 2015*; *Figure 8A*). *Wt* MEFs treated with Rho activator exhibited more and thicker bundles of actin filaments, with mesh-like structure, across the dorsal side of the cells, but maintained their overall polarized morphology. This observation is consistent with the known function of RhoA to induce central stress fiber formation in fibroblasts (*Katoh et al., 2001a*; *Katoh et al., 2001b*). In contrast, MLCK-CA expression increased peripheral stress fibers and noticeably fewer bundles of actin filaments in the central portion but maintained the classic polarized morphology. When combined (Rho activator treatment + MLCK-CA expression), the effect was increased, with the cells' thick actin bundles along the periphery. These cells, however, retained their protrusive structures and polarized shape (*Figure 8A, D, and E*).

To our surprise, when we attempted to image the myosin dynamics in *wt* and *Mfn2*-null MEFs, we noticed a large percentage of *wt* MEFs with round shapes and peripherally enriched actin filaments after overexpressing the GFP-tagged myosin regulatory light chain (MRLC-GFP) (*Figure 8B and E*). Cells expressing MRLC-GFP were significantly smaller and became round (*Figure 8C and D*). However, they still preserved their filopodia and other cell protrusions. When these cells were treated with the Rho activator, a noticeable percentage (24%) displayed the PAB morphology, including a decrease in cell area and an increase in cell roundness (*Figure 8B–F*). In summary, the combination of MRLC overexpression with pharmacological activation of RhoA is sufficient to drive PAB formation in MEFs.

## Loss of MFN2 displays different cytoskeletal architecture with increased cell contractile forces

To better virtualize the individual actin filaments in the PAB, we performed 3D super-resolution imaging of F-actin in *wt* and *Mfn2*-null MEFs (*Figure 9A*). *Wt* MEF displays interconnected actin network including peripheral actin fibers (blue box) parallel to the cell membrane and branched actin filaments pointing to cell protrusions against membrane (green box). However, in *Mfn2*-null MEFs,

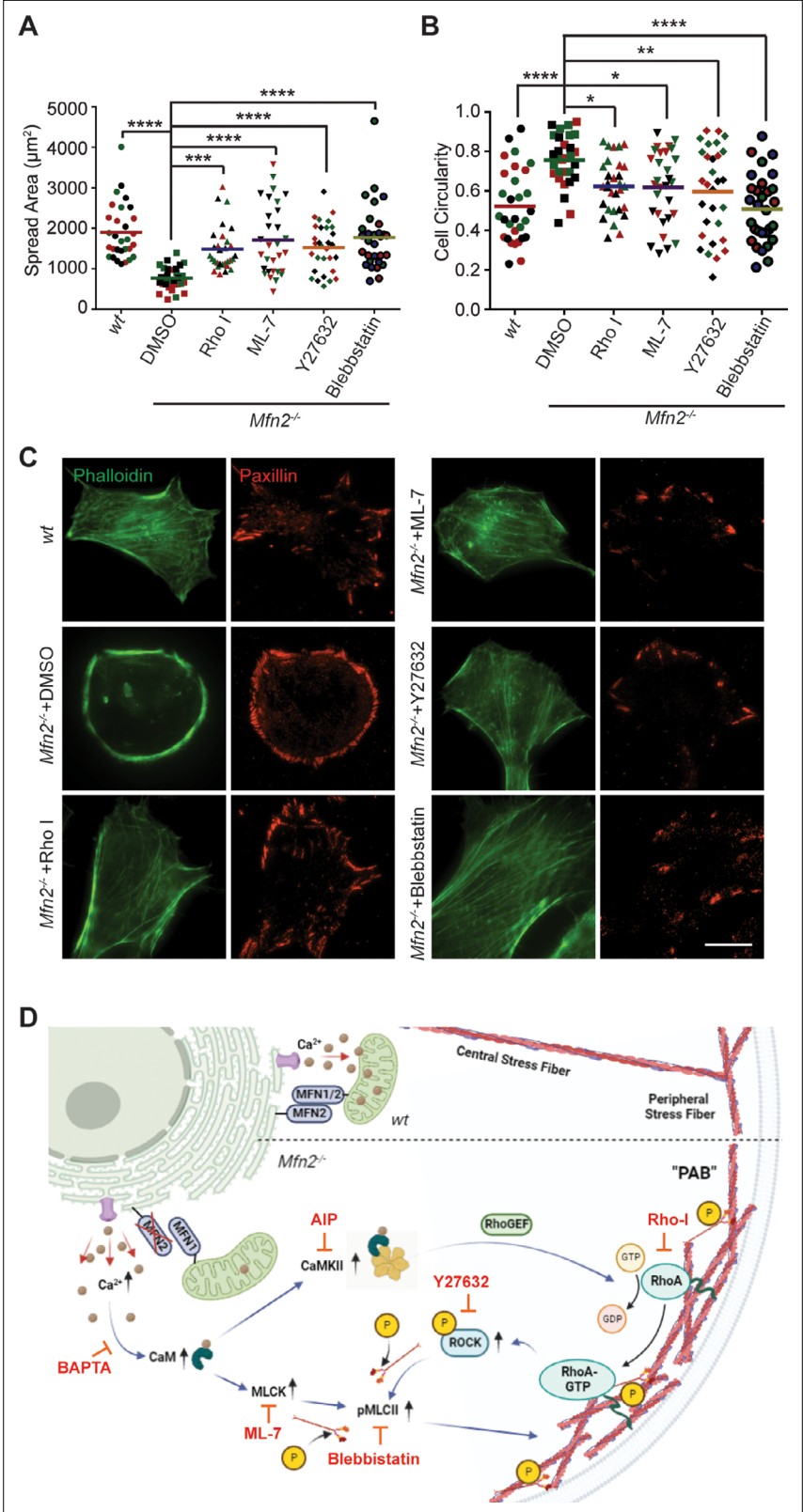

**Figure 6.** Small-molecule inhibitors targeting RhoA- and MLCK-related signaling pathways rescue MFN2 deficiency-induced phenotypes. (**A, B**) Cellular spread area (**A**) and circularity (**B**) of *wt* and *Mfn2*-null mice embryonic fibroblasts (MEFs) treated with indicated inhibitors overnight. (**C**) Immunofluorescence of F-actin (phalloidin) and Paxillin in *wt* and *Mfn2*-null MEFs treated with indicated inhibitors overnight. (**D**) Schematic of

*Figure 6 continued on next page*

*Figure 6 continued*

the effectors and their inhibitors (red) in MFN2-regulated signaling network leading to Actin bundle. Increased cytosolic Ca²⁺ may activate MLCK, CaMKII, and RhoA-ROCK, which activate MLC and affect actin bundle formation. One representative result of three biological repeats is shown in (**C**). Data are pooled from three independent experiments, and n = 30 cells are quantified in (**A, B**). *p≤0.05, **p≤0.01, ***p≤0.001, ****p<0.0001 (one-way ANOVA comparing each group to the average of *Mfn2⁻/⁻* DMSO group). Scale bars: 50 µm.

The online version of this article includes the following figure supplement(s) for figure 6:

**Figure supplement 1.** RhoA and MLC drive peripheral actin band ('PAB') structure in MFN2-deficient mice embryonic fibroblasts (MEFs).

---

thick actin bundles are present parallel to the cell membrane (blue box) with a region of disordered meshwork attached to the membrane (green box). Overall, stress fibers at the cell center are largely missing in *Mfn2*-null MEFs. We then used atomic force microscopy (AFM) to measure cell stiffness. *Mfn2*-null MEFs showed a softer Young's modulus than *wt* cells (*Figure 9—figure supplement 1A–C*), consistent with the previous report that cells with apical stress fibers are stiffer than cells without (*Efremov et al., 2019*). We then measured plasma membrane tension using the Flipper-TR dye and FLIM imaging (*Colom et al., 2018*). Upon addition of the hyperosmotic sucrose solution, Flipper-TR lifetime dropped significantly in MEFs, confirming that a longer probe lifetime corresponds to a membrane environment under higher tension (*Figure 9—figure supplement 1D and E*). Consistently, *Mfn2*-null MEFs showed a lower fluorescence lifetime, indicating lower membrane tension than the *wt* (*Figure 9-figure supplement 1F and G*). The reduced membrane stiffness and tension are consistent with a less spread cell morphology.

The prominent peripheral F-actin bundled with MLC (*Figure 7E*) possibly generates a nonpolar, global contraction at the cell periphery. We, therefore, utilized traction force microscopy (TFM) to measure the contractile force of the *Mfn2*-null MEFs (*Oakes et al., 2014*). After normalizing to the cell area, the average strain energy in *Mfn2*-null MEFs is significantly higher than with *wt* cells, which correlated with the elevated actin-myosin level. Since fibroblasts generate more traction force on stiffer substrates (*Lo et al., 2000*), we then asked whether or to which extent substrate stiffness affects the PAB structure in *Mfn2*-null MEFs. We cultured the cells on polyacrylamide (PAA) gels with different stiffness and stained F-actin (*Figure 9F*). As previously reported, *wt* cells showed diminished cell spreading and rounder morphology on softer substrates. In contrast, *Mfn2*-null MEFs displayed a more elongated cell shape on softer substrates, with partially restored stress fibers at the cell center (*Figure 9F and G*). These results suggest that substrate stiffness affects cell spreading, and strong substrate interaction or outside-in signal is required for the 'PAB' structure in *Mfn2*-null MEFs.

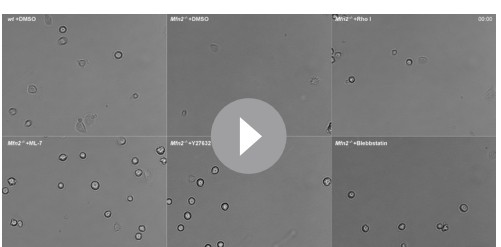

**Video 7.** Small-molecule inhibitors targeting RhoA and MLCK downstream signaling pathways rescue the migration defects in *Mfn2*-null mice embryonic fibroblast (MEF) cells. Cell spreading and random migration of *wt* treated with DMSO and *Mfn2*-null MEFs treated with DMSO, RhoA inhibitor-I (0.1 µg/ml), ML-7 (2 µM), Y29632 (5 µM), or Blebbistatin (4 µM) in the µ-slide. Time-lapse images were taken every 10 min for 14 hr and 30 min. MEFs were tracked for velocity quantification. Scale bar: 50 µm.

https://elifesciences.org/articles/88828/figures#video7

## Discussion

Our results show that loss of MFN2 protein results in defective spreading and polarization, along with reduced motility of MEF cells in 2D migration. This phenotype results from an increased cytosolic Ca²⁺ level upon Mfn2 removal and loss of ER-mitochondria tethers, which lead to the higher activity of calcium-regulated kinases, including CaMKII and MLCK, overactive RhoA and NMII, and an accumulation of peripherally localized actin and myosin, which we named the 'PAB.' The cell morphology of *Mfn2*-null MEFs can be rescued by restoring MFN2 expression, introducing ER-mitochondria artificial tethers, or inhibiting cytosolic Ca²⁺, CaMKII, MLCK, RhoA, or MRLC. Thus, together, these data identify the mechanism for how Mfn2 regulates cell spreading

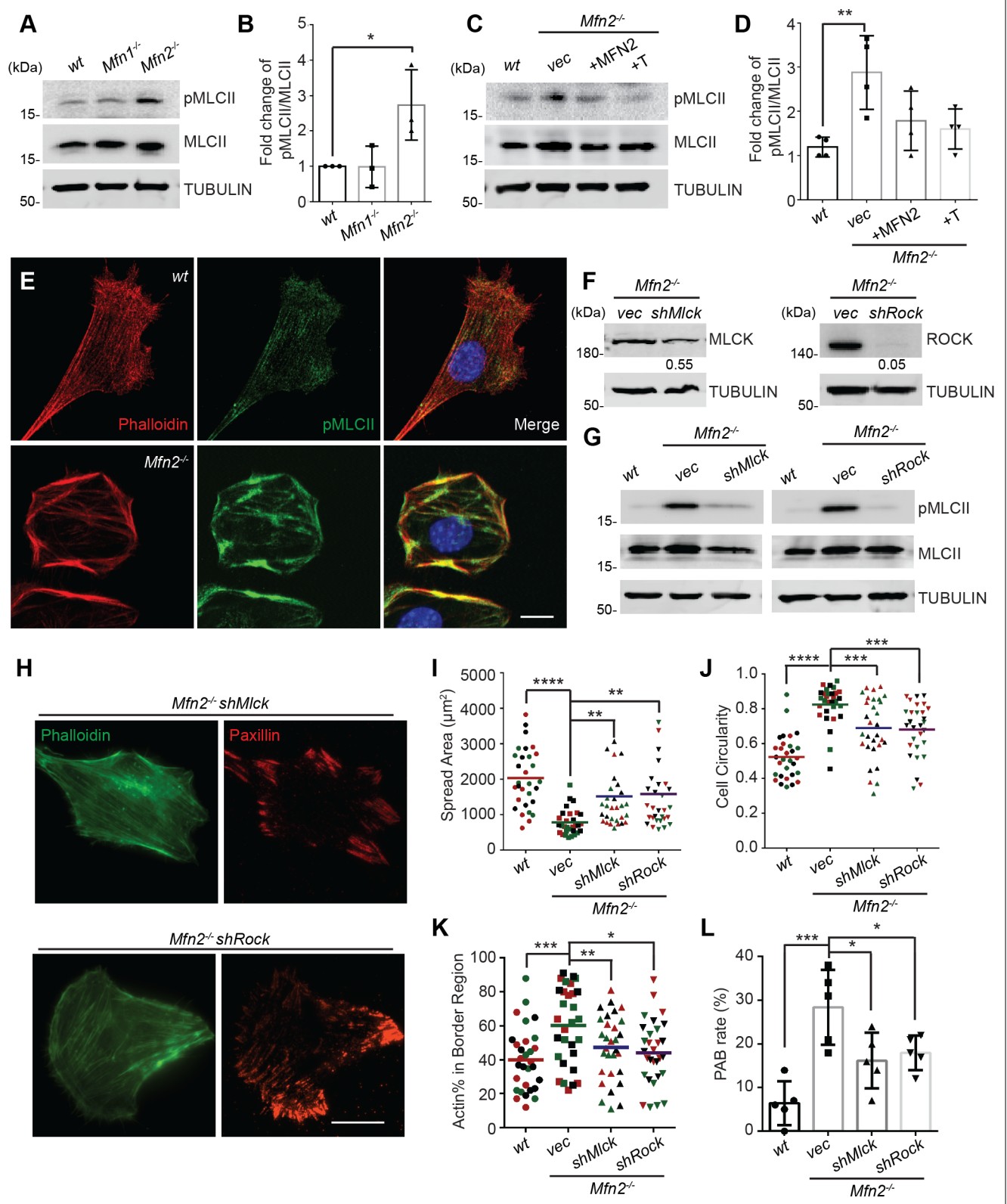

**Figure 7.** Heightened MLC activity promotes the peripheral actin band ('PAB') structure in MFN2-deficient mice embryonic fibroblasts (MEFs). (**A, B**) Western blot (**A**) and quantification (**B**) of the amount of pMLCII and total MLCII in *wt*, *Mfn1*-null, and *Mfn2*-null MEFs. (**C, D**) Increased pMLCII in *Mfn2*-null MEFs can be corrected by re-expressing MFN2 or inducing a mitochondria-endoplasmic reticulum (ER) tether. (**C**) Western blot and (**D**) quantification determining the amount of pMLCII and total MLCII protein in *wt*, *Mfn2*-null MEFs, *Mfn2*-null MEFs with MFN2 re-expressed, or with

*Figure 7 continued on next page*

*Figure 7 continued*

an artificial ER-mitochondria tether. (**E**) Representative images of *wt* and *Mfn2*-null MEFs immunostained for F-actin (phalloidin), pMLCII, and DAPI. (**F**) Western blot determining the expression levels of MLCK or ROCK in *Mfn2*-null MEFs with *shMLCK* or *shROCK*. (**G**) Western blot of pMLCII and total MLCII *Mfn2*-null MEFs with *shMLCK* or *shROCK*. (**H**) Representative images of *Mfn2*-null MEFs with *shMLCK* or *shROCK* immunostained for F-actin (green) and paxillin (red). (**I, J**) Cellular spread area and circularity of *wt, Mfn2*-null MEFs with *vec, shMLCK,* or *shROCK* were measured after overnight culture. (**K**) Percentage of actin abundance in the cell border region in *wt, Mfn2*-null MEFs, *Mfn2*-null MEFs with *shMLCK* or *shROCK*. (**L**) Percentage of PAB cells identified by a custom algorithm in *wt, Mfn2*-null MEFs, *Mfn2*-null MEFs with *shMLCK* or *shROCK*. The individual points stand for the size or circularity of individual MEF cells. One representative result of three biological repeats is shown in (**A, B, F, G**). Four biological repeats were done in (**C, D**). Data are pooled from three independent experiments in (**I, J**). n = 30 cells are quantified in (**I, K**). Five different views from three biological repeats are quantified in (**L**). *$p \leq 0.05$, **$p \leq 0.01$, ***$p \leq 0.001$, ****$p < 0.0001$ (one-way ANOVA, comparing each group to the average of *Mfn2$^{-/-}$ vec* group in **I, K**). Scale bars: 20 μm in (**H**), 10 μm in (**E**).

The online version of this article includes the following source data and figure supplement(s) for figure 7:

**Source data 1.** Original blots and figures with the bands labeled for *Figure 7A*.

**Source data 2.** Original blots and figures with the bands labeled for *Figure 7C*.

**Source data 3.** Original blots and figures with the bands labeled for *Figure 7F*.

**Source data 4.** Original blots and figures with the bands labeled for *Figure 7G*.

**Figure supplement 1.** The cell lines with restored motility show different oxygen consumption rates.

and adhesive migration in MEFs and highlight its essential function in maintaining mitochondria-ER contact and regulating the actomyosin network.

In line with the 'PAB' structure and markedly enriched myosin at the cell periphery in *Mfn2*-null MEFs, they displayed higher strain energy as determined by TFM. The findings suggest that increased peripheral traction force hampered *Mfn2*-null MEFs from further spreading. However, the MFN2 KO cells have increased actomyosin contractility only at the cell–substrate interface but not throughout the entire cell cortex. A less spread cell would have a more relaxed membrane and display a lower membrane tension, consistent with our observation in FLIM imaging of Flipper-TR dye. Softer matrices reduce cell contractility at the cell–substrate interface, which allows MFN2 KO cells to relax and spread better, given that a certain percentage of *Mfn2*-null MEFs gained elongated morphology on soft substrates. It is well known that mechanical forces play a significant role in regulating cell adhesion and cytoskeletal organization (*Discher et al., 2005*; *Burridge and Chrzanowska-Wodnicka, 1996*). Cells generate higher traction stresses on stiffer substrates, and contractility-induced tension drives the formation of stress fibers and focal adhesions. Conversely, stress fibers and focal adhesions disassemble when contractility is inhibited (*Burridge and Chrzanowska-Wodnicka, 1996*; *Pelham and Wang, 1997*; *Wong et al., 2015*). A possible explanation of our observation is that the 'outside-in' feedback loop coupling with the elasticity of the extracellular microenvironment abrogated the aberrant 'PAB' architecture.

Cell migration is a highly dynamic process in which actin treadmilling and focal adhesion turnover orchestrate front-to-rear polarity and cell movements. Rac is known for its regulatory functions in the formation of focal complexes and lamellipodia at the front (*Hall, 2005*; *Nobes and Hall, 1999*; *Nobes and Hall, 1995*), while RhoA modulates actomyosin contraction (*Ridley and Hall, 1992*) and focal adhesion disassembly (*Ridley and Hall, 1992*; *Nobes and Hall, 1999*; *Nobes and Hall, 1995*) at the rear end. Our previous research proved that MFN2 suppresses Rac activation and supports neutrophil cell adhesion (*Zhou et al., 2020*). However, we found that RhoA, instead of Rac, dominates actin cytoskeleton reorganization in *Mfn2*-null MEFs. This difference is not entirely surprising, given that these two cells utilize different migratory modes. Neutrophils are a type of fast amoeboid-migrating cell that do not form mature focal adhesions during cell migration, even on specific substrates (*Lämmermann et al., 2008*; *De Bruyn, 1946*). In contrast, slower-moving cells, like fibroblasts, form mature focal adhesions and require focal adhesion recycling to move (*Lauffenburger and Horwitz, 1996*; *Seetharaman and Etienne-Manneville, 2020*). Despite described controversies concerning the dominant downstream effectors in neutrophils and fibroblasts, the proposed negative regulatory function of MFN2 on cytosolic $Ca^{2+}$ levels and its functional role in cell migration was consistent in both cell lines.

The elevation of cytosolic $Ca^{2+}$ concentration is known to induce the activation of MLCK. MLCK phosphorylates the 20 kDa regulatory MLC at S19 and consequently activates the myosin ATPase activity (*Ikebe and Hartshorne, 1985*). The Rho-kinase, ROCK, also controls MLC activity by phosphorylating at Ser19 and Thr 18 (*Kassianidou et al., 2017*; *Totsukawa et al., 2000*). It is reported

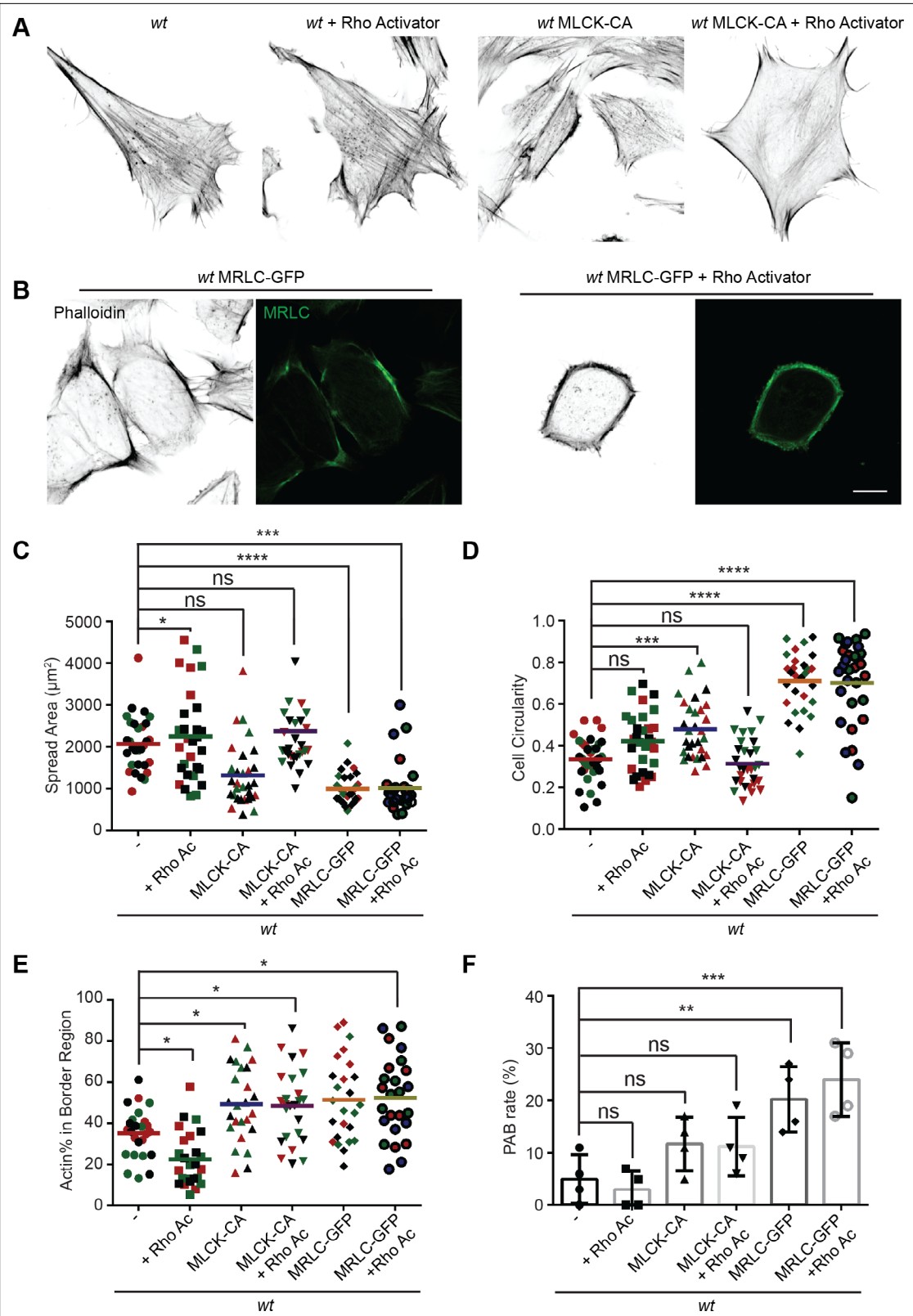

**Figure 8.** MRLC overexpression and Rho activation in *wt* mice embryonic fibroblasts (MEFs) recapitulates the peripheral actin band ('PAB') structure. (**A**) Immunofluorescence of F-actin (phalloidin) in *wt* MEFs with or Rho activator treatment or introducing MLCK-CA expression. (**B**) Representative images of *wt* MEFs expressing MRLC-GFP with or without the Rho Activator, immunostained for F-actin (phalloidin). (**C, D**) Spread area and circularity of *wt* MEFs with the indicated overexpression or drug treatment. The individual points stand for the size or circularity of individual MEF cells. (**E,**

*Figure 8 continued on next page*

*Figure 8 continued*

**F**) Percentage of actin abundance in the cell border region (**E**) and the percentage of PAB cells (**F**) identified by our custom algorism in indicated cell lines. One representative result of three biological repeats is shown in (**A, B**). Data are pooled from three independent experiments in (**C–E**). n = 30 cells are tracked and counted in (**C–E**). Four different views from three biological repeats are quantified in (**F**). *p≤0.05, **p≤0.01, ***p≤0.001, ****p<0.0001 (one-way ANOVA, comparing each column with the mean of the *wt* group). Scale bars: 10 µm.

that the biphosphorylated MLC (pp-MLC) localizes to the cell center. In contrast, the monophosphorylated MLC (p-MLC) tends to be located in the cell periphery (*Kassianidou et al., 2017*). Indeed, this is consistent with our observation that the pp-MLC level did not change when we probed pp-MLC with an antibody specific for both Ser19 and Thr18 (data not shown). Only MLC pSer19 was markedly elevated in *Mfn2*-null MEFs. However, interestingly, either MLCK inhibitor ML-7 or ROCK inhibitor Y27632 restored cell morphology and motility in MEFs without MFN2. Our data suggested that both ROCK and MLCK are required for enhanced MLC phosphorylation at Ser19 in the MFN2-null MEFs.

The RhoA/ROCK signaling pathway is essential in response to cytosolic $Ca^{2+}$ (*Uehata et al., 1997*; *Saneyoshi and Hayashi, 2012*; *Ying et al., 2009*). PDZ-RhoGEF is a vital effector in response to cytosolic $Ca^{2+}$ (*Ying et al., 2009*; *Derewenda et al., 2004*). Cytosolic $Ca^{2+}$ activates RhoA through the PYK2/PDZ-RhoGEF pathway in five cell lines (Primary rat aortic vascular smooth muscle cells, HEK293T, MDCK, Neuron2A, and PC12) (*Ying et al., 2009*). In line with the previous observation (*Ying et al., 2009*), BAPTA-AM abolishes RhoA activation in our current work. Further work will be required to determine whether PDZ-RhoGEF or other molecules mediate Rho activation with heightened intracellular $Ca^{2+}$ levels in fibroblasts.

Besides the effectors we identified here, focal adhesion proteins, including focal adhesion kinase (FAK) and proteins of the FAK–Src signaling complex, are also known as crucial modulators participating in interactions with the extracellular matrix and the cytoskeleton (*Parsons et al., 2010*; *Giannone et al., 2007*; *Gardel et al., 2010*; *Mitra et al., 2005*). Intracellular forces generated by focal adhesion proteins promote rear retraction and the forward movement of the cell. The dynamic turnover of focal adhesions is spatiotemporally controlled by intracellular $Ca^{2+}$ signaling (*Machacek et al., 2009*; *Mitra et al., 2005*). In addition, calpains allow the degradation of FAPs, including FAK and Talin, in a $Ca^{2+}$-dependent manner (*Kerstein et al., 2017*; *Goll et al., 2003*). Based on the extensive peripheral focal adhesions we observed in *Mfn2*-null MEFs, calpains or other focal adhesion proteins may also contribute to the 'PAB' structure.

Neonates are susceptible to MFN2 defects (*Filadi et al., 2018*), and over 100 dominant mutations in the MFN2 gene have been reported in Charcot–Marie–Tooth disease type 2A (CMT2A) patients. However, how these mutations lead to disease is largely unknown, and there is currently no cure for this disease (*Verhoeven et al., 2006*; *Calvo et al., 2009*). MFN2 mutations are also associated with many other conditions, such as Alzheimer's disease, Parkinson's disease, obesity, and diabetes (*Kim et al., 2017*; *Wang et al., 2009*; *Lee et al., 2012*; *Bach et al., 2003*). One of the challenges in MFN2 research is that MFN2 plays multiple functional roles in cell signalings, such as regulating mitochondrial dynamics, transport, mtDNA stability, lipid metabolism, and cell survival. Both gain-of-function and loss-of-function mutations are reported in CMT2A patients. Some MFN2 mutations lead to fusion-incompetent mitochondria. However, some are fusion-competent mutations (*Strickland et al., 2014*; *Cartoni et al., 2010*; *El Fissi et al., 2018*; *Franco et al., 2016*; *Rocha et al., 2018*). It is also worth noting that MFN2 dysfunction preferentially impacts peripheral nerves instead of central nerves. These phenotypes are generally attributed to deficient mitochondrial trafficking and localization to the dendrites (*Pareyson et al., 2015*; *Baloh et al., 2007*). Our observations may provide a potential novel explanation of how MFN2 deficiency affects cell physiology by modulating the mechanosensitivity and cytoskeletal organization of the cells. It is possible that the MFN2 disease mutations also disrupt the mitochondria-ER tether and result in defects in cell cytoskeleton architecture, cell spreading, and migration, which cause the progression of the diseases. It is also possible that the same MFN2 mutation induces distinct signaling alterations in different cell types or diseases.

In summary, we characterized the alteration of the cytoskeleton and biophysical properties in MFN2-deficient cells and identified the detailed molecular mechanism. Our work provides insights into how MFN2 deficiency affects cell morphology and motility, specifically via the actin cytoskeleton.

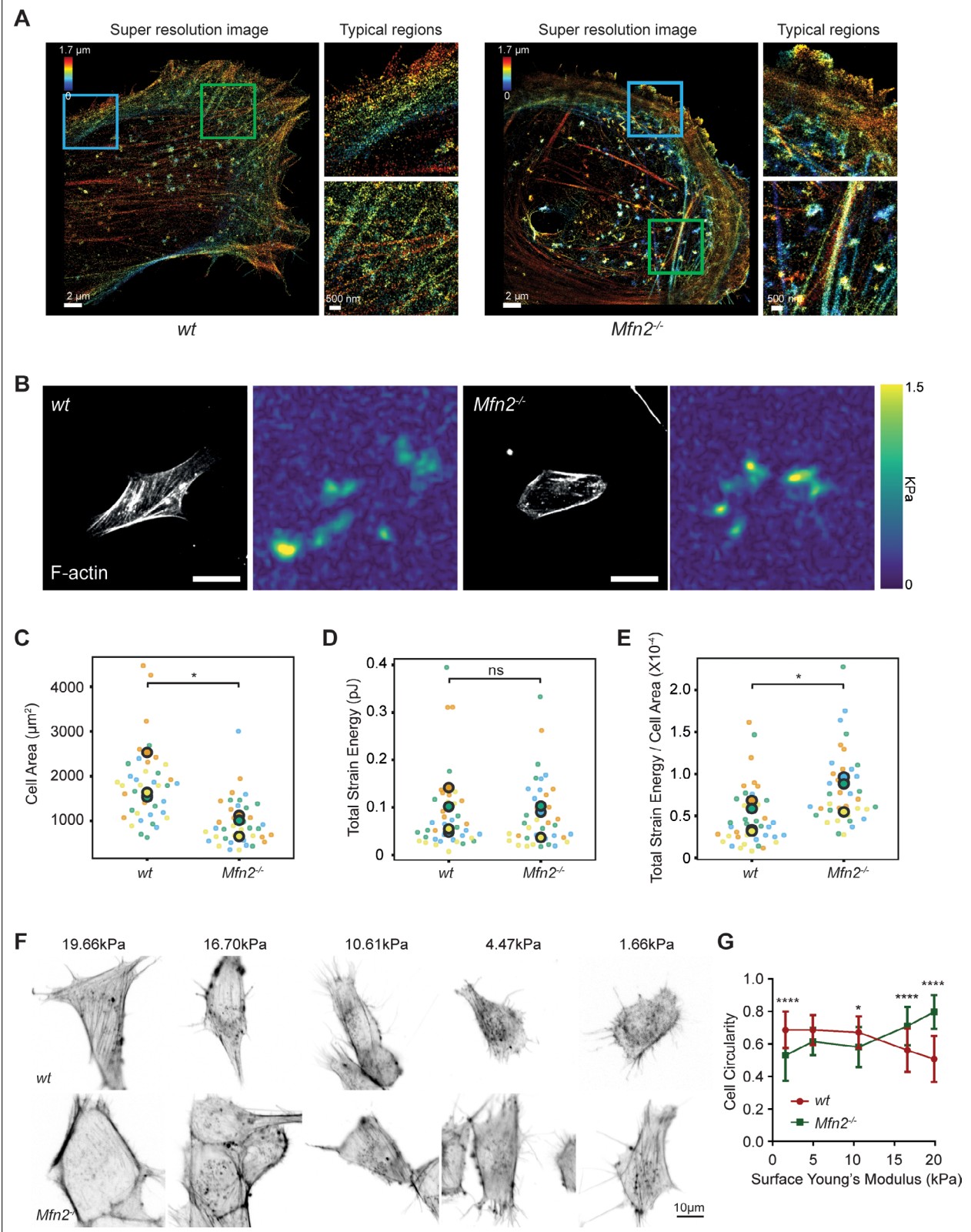

**Figure 9.** Mfn2-null mice embryonic fibroblasts (MEFs) exhibit altered actin organization and cell stiffness. (**A**) 3D super-resolution reconstructions of immunofluorescence-labeled F-actin in wt or Mfn2-null MEFs. x–y overview of a 1.7-µm-thick volume of the cells. (**B**) Morphology of indicated MEF cells on polyacrylamide (PAA) substrates after overnight culture immunostained with Alexa-488 phalloidin. A traction stress map with color values corresponding to different stress values is shown on the right. Scale bars: 50 µm. Quantification of the corresponding cell spreading areas (**C**), total

*Figure 9 continued on next page*

Figure 9 continued

strain energy (**D**), and total strain energy normalized to cell area (**E**). (**F**) Representative images of indicated MEF cells on polyacrylamide (PAA) substrates of different stiffness after overnight culture. The cells are immunostained with Alexa-488 phalloidin. (**G**) The cell circularity of the indicated cells is measured. The individual points stand for the circularity of individual MEF cells. One representative result of three biological repeats is shown in (**C, F**). Data are pooled from three independent experiments in (**C–E, G**). n > 40 cells are counted in (**B–E**). *p≤0.05, ****p<0.0001 (unpaired *t*-test). Scale bars: 2 μm in (**A**), 50 μm in (**B**), and 10 μm in (**F**).

The online version of this article includes the following figure supplement(s) for figure 9:

**Figure supplement 1.** *Mfn2*-null mice embryonic fibroblasts (MEFs) exhibit altered cell stiffness and membrane tension.

## Materials and methods

### Cell culture

HEK293T (CRL-11268), wild-type (CRL-2991), *Mfn2*-null (CRL-2993), and *Mfn1*-null (CRL-2992) MEFs were from the American Type Culture Collection (ATCC, Manassas, VA). GP2-293 cells were purchased from Takara Bio USA (#631458). All cells were maintained at 37°C with 5% $CO_2$ in a Forma Steri-Cycle i160 CO2 Incubator (NC1207547, Thermo Fisher Scientific). Cells were cultured in 10% FBS in DMEM with sodium bicarbonate. Cell lines were authenticated using STR profiling. Mycoplasma contamination was absent in monthly tests using Mycoplasma Detection PCR. To obtain MFN2 re-expression cell line, introduce the ER-mito tethering structure, or express CaMKII-WT, CaMKII-DN, MLCK-CA, or MRLC-GFP, the MSCV-puro vector (Takara Bio USA, #634401) were used. We cloned the gene of interest into the MSCV vector and cotransfected the vector plasmid along with the envelope plasmid pVSVG, at a ratio of 1:1, into GP2-293 cells using Lipofectamine 3000 (Invitrogen L3000015). Virus supernatant was collected at 48 hpt and 72 hpt and further concentrated with Lenti-X concentrator (Clontech 631232). MEF cells were transduced with concentrated retrovirus in a complete medium and then selected with 2 μg/ml puromycin (Gibco A1113803) starting the next day. The stable lines were generated after puromycin selection for 1 wk. To generate ROCK or MLCK knocking down lines in *Mfn2-null* MEF cells, pLKO.1 lentiviral constructs with shRNAs were obtained from Sigma-Aldrich (shROCK: TRCN0000022903, shMLCK: TRCN0000024037), and SHC 003 was used as a non-targeting control. The plasmid of lentiviral constructs together with pCMV-dR8.2 dvpr (Addgene #8455) and pCMV-VSV-G (Addgene #8454), at a ratio of 10:7.5:2.5, were cotransfected into HEK293T cells with Lipofectamine 3000 (Invitrogen L3000015) to produce lentivirus. Virus supernatant was collected at both 48 hpt and 72 hpt, and further concentrated with Lenti-X concentrator (Clontech 631232) before transduction. 2 μg/ml puromycin (Gibco A1113803) was added into the complete medium on the next day for selection.

### Plasmids

In-Fusion cloning (In-Fusion HD Cloning Plus Kit, Clontech) was used to fuse the fragments with the linearized backbone. MSCV-puro was digested by Xho I and EcoR I. pLIX_403 plasmid was digested by NsiI and BamHI. The plasmids of Mfn1-Myc and Mfn2-Myc were gifts from David Chan (Addgene plasmid # 23212, #23213). The mito-GFP-ER plasmid was from the plasmid used in our lab (Addgene plasmid #160509). pLIX_403 was a gift from David Root (Addgene plasmid # 41395). GFP-C1-CAMKIIalpha and GFP-C1-CAMKIIalpha-K42R were gifts from Tobias Meyer (Addgene #21226, #21221). pTK91_GFP-MRLC2 was also from Addgene (Addgene #46355). The lentiviral backbone pLIX_403 was a gift from David Root (Addgene plasmid # 41395). pSLIK CA MLCK was a gift from Sanjay Kumar (Addgene plasmid # 84647). pYFP-paxillin was a gift from Kenneth Yamada (Addgene #50543). SPLICS Mt-ER Long P2A and pCytERM_mScarlet_N1 were gifts from Dorus Gadella (Addgene plasmid #164107 and # 85066).

The In-Fusion primers are listed below:

MSCV-mfn2 insert F: CACGATAATACCATGGGCCACCATGTCCCTGCTC
MSCV-mfn2 insert R: TCTAGAGTCGCGGCCGCTTACTTGTACAGCTCGTCCATGCC
MSCV-mfn1 insert R: TCGACTCTAGAGTCGCGGCCGCTTACTTGTACAGCTCGTCCATGCC
Mfn2 into plix-Nsil-F: AAAACCCCGGTCCTATGCATATGTCCCTGCTCTTCTCTCGA
Mfn2 into plix-BamHI-R: CCCCAACCCCGGATCCTTATCTGCTGGGCTGCAGGT
Camk2a-MSCV-F: AATTAGATCTCTCGAGGCCACCATGGTGAGCAAGG
Camk2a-MSCV-R: CTACCCGGTAGAATTCATTCGGCGAAGCAAGAGCG

ER-mito F: AATTAGATCTCTCGAGATGGCAATCCAGTTGCGTTCG
ER-mito R: ATTTACGTAGCGGCCGCTTAAGATACATTGATGAGTTTGG
MRLC-GFP F: AATTAGATCTCTCGAGGCCACCATGGTGAGCAAGG
MRLC-GFP R: CTACCCGGTAGAATTCGCCCGCGGTCAGTCATCTTTG
MLCK-CA F: attagatctctcgagactagtcgactggatcc
MLCK-CA R: ccggtagaattcagatcttgggtgggttaattaa

## Chemicals

MEF cells were treated with 1% DMSO, 20 µM BAPTA (Cayman Chemical), Y27632 5 µM (Cayman Chemical), 50 µM CK666 (Cayman Chemical), STO-609 acetate (Biotechne, #1551), A23187 (Cayman Chemical), 4 µM Blebbistatin (Cayman Chemical), 40 uM AIP (R&D Systems #5959/1), 50 µM CAS 1090893 (Millipore, #553511), 0.1 µg/ml RhoA inhibitor-I (Cytoskeleton, Inc, #CT-04), 2 µM FK-506 (Cayman Chemical, # 10007965), 300 nM FAK14 (Cayman Chemical, #14485), 10 µM BMS-5 (Cayman Chemical, #21072), or 2 µM ML-7 (Cayman Chemical, #11801) overnight during time-lapse imaging for random cell migration or followed by immunostaining.

## Western blot

Total protein was isolated from cells using RIPA buffer containing 25 mM Tris-HCl (pH 8.0), 150 mM NaCl, 1 mM EDTA, 0.5% NP-40, 0.5% sodium deoxycholate, and 0.1% sodium dodecyl sulfate (SDS). For samples containing phosphor proteins to probe, 20 mM sodium fluoride (NaF), 1 mM sodium orthovanadate ($Na_2VO_3$), 10 mM beta glycerophosphate were added to the RIPA lysis buffer. Protein concentrations were determined using the Precision Red Advanced Protein Assay Reagent (Cytoskeleton ADV02). Extracted proteins (25–35 µg) were separated by 8–12% SDS-PAGE and transferred onto polyvinylidene difluoride membranes (PVDF, Bio-Rad). Membranes were blocked for ~30 min in PBST (PBS and 0.1% Tween 20) with 5% fat-free milk. After blocking, membranes were incubated with primary antibodies diluted 1:1000 in 1% BSA at 4°C overnight and secondary antibodies diluted 1:10,000 in PBST at room temperature for 1 hr. Odyssey (LI-Cor) was used to image membranes. Immunodetection of the pulldown samples was performed using enhanced Western Blotting Chemiluminescence Luminol Reagent (Santa Cruz Biotechnology, Cat# sc-2048) and detected with a FluorChem R System (Proteinsimple). Image Studio 5.0 was used to quantify and analyze the results. Primary antibodies anti-Mfn2 (Cell Signaling 9482S), anti-Mfn1 (Abcam, ab126575), anti-pan-CaMKII (Cell Signaling #3362), anti-phosphor-CaMKII (Thr286) (Cell Signaling #12716), anti-phospho myosin light chain 2 (Cell Signaling #3671), anti-myosin light chain 2 (Cell Signaling #3672), anti-phospho-PAK (Cell Signaling #2605S), anti-PAK (Cell Signaling #2604), and secondary antibody HRP AffiniPure goat anti-rabbit IgG (Jackson ImmunoResearch, #111-035-003), goat anti-mouse IgG Alexa Fluor 680 (Invitrogen, #A28183), and goat anti-rabbit IgG Alexa Fluor Plus 800 (Invitrogen, #A32735).

## Rac-GTP and RhoA-GTP pulldown assay

PAK-GST-coated beads (Cytoskeleton BK035) and Rhotekin-RBD-coated beads (Cytoskeleton BK036) were used to isolate active Rac and RhoA from the whole-cell lysate. MEF cells were serum starved with DMEM medium lacking FBS overnight in the incubator at 70–80% confluency. After starvation, PDGF-BB was then added to the cells at a final concentration of 25 µM. Then, cells were lysed with ice-cold lysis buffer at indicated time points and collected by scrapples. Then, 15 µg PAK-GST beads or 50 µg Rhotekin-RBD-coated beads were mixed with each sample and incubated at 4°C for 1 hr. Protein beads were washed and processed for western blot.

## Immunostaining and confocal imaging

For immunofluorescent staining, MEF cells with or without drug treated overnight were plated onto coverslips and incubated overnight at 37°C, then fixed with 4% paraformaldehyde (PFA) solution in PBS for 15 min at room temperature. Cells were permeabilized in PBS with 0.1% Triton X-100 and 3% fatty acid-free BSA for 1 hr, then incubated with phalloidin Alexa Fluor 488 (Invitrogen A12379) or primary antibodies diluted 1:100 in 3% BSA overnight at 4°C. After washing with PBS three times, the cells were stained with secondary antibodies diluted 1:500 in 3% BSA and DAPI (Invitrogen D3571) for 1 hr at room temperature. After washing with PBS three times, the coverslips were mounted on glass slides with the mounting medium (Vector Laboratories H-1000). Primary antibodies anti-Mfn2

(Cell Signaling 9482S), anti-paxillin (Invitrogen AHO0492), and anti-phospho myosin light chain 2 (Cell Signaling 3671), and secondary antibodies anti-rabbit Alexa Fluor 488 (Invitrogen A-21441) and anti-mouse Alexa Fluor 568 (Invitrogen A-11004) were used. Images of F-actin and focal adhesions were acquired using an N-STORM/N-SIM TIRF microscope (Nikon) with a 1.49/60× Apo TIRF oil objective. For focal adhesion quantification, the built-in 'threshold' plugin in ImageJ was first used to isolate the focal adhesions in the images. Then the 'analysize particles' plugin was used to quantify the numbers and size. Images of F-actin with MFN2 or pMLCII were acquired by a laser-scanning confocal microscope (LSM 800, Zeiss) with a 1.4/63× oil immersion objective lens. Images were processed and analyzed with ImageJ. To quantify the cell area and circularity, phalloidin-stained cells were imaged with a 40× objective. Images were imported into ImageJ, FiloQuant plugin was used to identify cell edges and cytoskeleton, and 'Particle Analysis' was then used to calculate the cell area and circularity. A custom algorithm was developed to identify cells with PAB structure. The algorithm calculates the percentage of cytoskeleton intensity in the cell border region based on the output of the images by the FiloQuant plugin (available on GitHub, copy archived at *Tomato990, 2023*). SPLICS-L probe was used as described (*Vallese et al., 2020*), and the images of ER-mitochondrial contacts were acquired with a Nikon Ti2 Inverted Microscope with Yokogawa W1 and SoRa Module, 63× objective was used here. A 3D reconstruction of the resulting image was obtained using the Volume J plugin (available here). A selected face of the 3D rendering was then thresholded and used to count ER–mitochondria contact sites. Data were plotted in Prism 6.0 (GraphPad).

## 2D migration live imaging

MEF cells were first trypsinized and replated onto fibrinogen-coated μ-slide 8-well plates (ibidi 80826) at a density of ~10,000 cells per well with a complete medium. Time-lapse images were acquired using BioTek Lionheart FX Automated Microscope with 20× phase lens at 10 min intervals of ~18 hr at 37°C with 5% $CO_2$. The velocity of MEFs was measured using ImageJ with the MTrackJ plugin and plotted in Prism 6.0 (GraphPad). The rose plots and directionality index were generated or calculated by the Chemotaxis and Migration Tool 2.0.

## Ca²⁺ measurement

Fluo-4 Calcium Imaging Kit (Invitrogen F10489) was used for cytosolic $Ca^{2+}$ measurement in MEFs. MEF cells were incubated with PowerLoad solution and Fluo-4 dye at 37°C for 15 min and then at room temperature for 15 min. After incubation, cells were washed with PBS one time. Time-lapse green fluorescence images were obtained with AXIO Zoom V16 microscope (Zeiss) at 1 min intervals of 25 min. Then, 50 µl of 2 mM PDGF-BB (Sigma, #P4056) was added to cells right after the first image was taken. The fold change of the fluorescence intensity was normalized to that of the first image. The fluorescence intensity was measured using ImageJ and plotted in Prism 6.0 (GraphPad).

## 3D super-resolution imaging

Single-molecule super-resolution imaging was performed on a custom-built setup on an Olympus IX-73 microscope stand (Olympus America, IX-73) equipped with a 100×/1.35-NA silicone-oil-immersion objective lens (Olympus America, UPLSAPO100XS) and a PIFOC objective positioner (Physik Instrumente, ND72Z2LAQ). Samples were excited by a 642 nm laser (MPB Communications, 2RU-VFL-P-2000-642-B1R), which passed through an acoustic-optic tunable filter (AA Opto-electronic, AOTFnC-400.650-TN) for power modulation. The excitation light was focused on the pupil plane of the objective lens after passing through a filter cube holding a quadband dichroic mirror (Chroma, ZT405/488/561/647rpc). The fluorescent signal was magnified by relay lenses arranged in a 4f align-ment to a final magnification of ~54 and then split with a 50/50 non-polarizing beam splitter (Thorlabs, BS016). Two mirrors delivered the split fluorescent signals onto a 90° specialty mirror (Edmund Optics, 47-005), axially separated by 590 nm in the sample plane, and then projected on an sCMOS camera (Hamamatsu, Orca-Flash4.0v3) with an effective pixel size of 119 nm. A bandpass filter (Semrock, FF01-731/137-25) was placed before detection. The imaging system was controlled by custom-written LabVIEW (National Instruments) programs.

Before imaging, the coverslip with cells on top was placed on a custom-made holder. Then, 100 µl of imaging buffer (10% [w/v] glucose in 50 mM Tris, 50 mM NaCl, 10 mM β-mercaptoethylamine hydrochloride [M6500, Sigma-Aldrich], 50 mM 2-mercaptoethanol [M3148, Sigma-Aldrich], 2 mM

cyclooctatetraene [138924, Sigma-Aldrich], 2.5 mM protocatechuic acid [37580, Sigma-Aldrich], and 50 nM protocatechuate 3,4-dioxygenase [P8279, Sigma-Aldrich], pH 8.0) were added on top of the coverslip. Then another coverslip was placed on top of the imaging buffer. This coverslip sandwich was then sealed with two-component silicon dental glue (Dental-Produktions und Vertriebs GmbH, picodent twinsil speed 22).

Following the previous procedure (*Xu et al., 2020*), the sample was first excited with the 642 nm laser at a low intensity of ~50 W/cm$^2$ to find a region of interest. Before fluorescence imaging, bright-field images of this region were recorded over an axial range from –1 to +1 µm with a step size of 100 nm as reference images for focus stabilization. Single-molecule blinking data were then collected at a laser intensity of 2–6 kW/cm$^2$ and a frame rate of 50 Hz. Imaging was conducted for ~30 cycles with 2000 frames per cycle. Single-molecule localization was performed as described previously (*Xu et al., 2020*).

## Seahorse mitochondrial respiration analysis

Mitochondrial respiration was measured with Seahorse XFe24 Analyzer (Agilent Technologies) according to the manual of Seahorse XF Cell Mito Stress Test Kit (Agilent Technologies, Cat# 103015-100). Briefly, MEF cells were plated on the XF24 cell culture microplate at a density of 50,000 cells per well. The seahorse sensor cartridge was hydrated with calibrant in a non-CO$_2$ incubator at 37°C overnight 1 d before measurement. On the day of measurements, cells were washed once and incubated in Seahorse XF base medium (pH 7.4, Agilent Technologies, Cat# 103334-100) supplemented with 1 mM sodium pyruvate, 2 mM glutamine, and 5.5 mM glucose. Cells were equilibrated at 37°C in a non-CO$_2$ incubator for 1 hr. The oxygen consumption rate was monitored at the basal state and after sequential injection of the mitochondrial compounds oligomycin (1 µg/ml), FCCP (1 µM), and Rotenone/antimycin A (both 1 µM) to induce mitochondrial stress. All mitochondrial respiration rates were generated and automatically calculated by the Seahorse Wave software with normalization to the cellular protein contents. Cellular protein contents were determined by the sulforhodamine B (SRB) assay as described (*Vichai and Kirtikara, 2006*).

## Traction force microscopy and analysis

TFM was performed as described previously (*Huang et al., 2019*; *Sala and Oakes, 2021*; *Aratyn-Schaus et al., 2010*). Briefly, 22 × 30 mm #1.5 glass coverslips were activated by incubating with a 2% solution of 3-aminopropyltrimethyoxysilane (313255000, Acros Organics) diluted in isopropanol, followed by fixation in 1% glutaraldehyde (16360, Electron Microscopy Sciences) in ddH$_2$0. Polyacrylamide gels (shear modulus: 16 kPa – final concentrations of 12% acrylamide [1610140, Bio-Rad] and 0.15% bis-acrylamide [1610142, Bio-Rad]) were embedded with 0.04 µm fluorescent microspheres (F8789, Invitrogen) and ~6 mg/ml acryloyl-X, SE (6-((acryloyl)amino)hexanoic acid)-labeled fibronectin (A20770, Thermo Fisher Scientific; FC010, EMD Millipore), and polymerized on activated glass coverslips for 1 hr at room temperature. After polymerization, gels were rehydrated for 1 hr in deionized H$_2$O before seeding $1.0 \times 10^5$ cells on each gel in a 60 mm cell culture-treated Petri dish. Cells were allowed to spread overnight, and the next day SPY555-actin dye (SC202, Spirochrome) was added ~60 min before imaging. Images were taken of both the cells and underlying fluorescent beads. Following imaging, cells were removed from the gel by adding 0.025% SDS, and a reference image of the fluorescent beads in the unstrained gel was taken.

Analysis of traction forces was performed using code written in Python according to previously described approaches (*Sabass et al., 2008*). Prior to processing, the reference bead image was aligned to the bead image with the cell attached. Displacements in the beads were calculated using an optical flow algorithm in OpenCV (Open Source Computer Vision Library, available on GitHub) (*OpenCV contributors, 2022*; *Bradski and Kaehler, 2008*) with a window size of 16 pixels. Traction stresses were calculated using the FTTC approach (*Butler et al., 2002*; *Huang et al., 2019*) as previously described, with a regularization parameter of $9.34 \times 10^{-9}$. The strain energy was calculated by summing one-half the product of the strain and traction vectors in the region under the cell (*Oakes et al., 2014*) and normalized by the cell area as measured using the SPY555-actin image of the cell. Cells with residual energy of ≥20% were excluded from the data set. N = 4 with ≥9 cells per biological repeat.

## Substrate stiffness assay

Polyacrylamide gels with uniform stiffness were prepared as previously described (*Efremov et al., 2022*). Briefly, 50 mm glass-bottom dishes (WPI, USA) were activated with 0.1 M NaOH, 4% (v/v) APTES ((3-aminopropyl)triethoxysilane, Sigma-Aldrich, USA) and 1% (v/v) glutaraldehyde (Sigma-Aldrich) in 1× PBS. To prepare a gel with a specific stiffness, different concentrations of acrylamide (40%, Sigma-Aldrich) and bis-acrylamide (1%, Sigma-Aldrich) were mixed with PBS 1× and 0.5% (w/v) Irgacure 2959 (0.5% w/v, (2-hydroxy-4'-(2-hydroxyethoxy))-2-methylpropiophenone, Sigma-Aldrich). Later, the gel solution was incubated at 37°C overnight and degassed for 30 min at room temperature. Next, to prepare one gel, 120 ul of the gel solution was poured into the center of an activated glass bottom dish and covered with 22 * 22 mm coverglass previously chloro-silanated with DCDMS (dichlorodimethylsilane, Sigma-Aldrich). Then, the dish was placed in a UV transilluminator for 10 min. Final concentrations of acrylamide (4%) and bis-acrylamide (0.2%, w/v) were chosen to prepare PAA gels with 1.67, 4.47, 10.61, 16.7, and 19.66 kPa Young's modulus. The ratio is shown below:

| Elastic modulus (kPa) | Acrylamide (ml) from 40% stoch solution | bis-Acrylamide (ml) from 2% stock solution | PBS 1× (ml) |
|---|---|---|---|
| 1.67 | 0.075 | 0.1125 | 0.8125 |
| 4.47 | 0.125 | 0.075 | 0.8 |
| 10.61 | 0.250 | 0.050 | 0.7 |
| 16.7 | 0.250 | 0.075 | 0.675 |
| 19.66 | 0.2 | 0.132 | 0.668 |

## Atomic force microscopy

An Asylum Research MFP3d Bio AFM system (Santa Barbara, CA) was used to measure the effective modulus of *wt* and *Mfn2*-null MEFs seeded in 60 mm polystyrene Petri dishes. A Nanoandmore CP-qp-SCONT-SiO-B-5 colloidal probe cantilever (Watsonville, CA) with a 3.5 µm probe diameter and 0.01 N/m nominal stiffness was used for these experiments. The optical lever sensitivity of the cantilever was calibrated using a static force–displacement curve, performed on the part of the polystyrene Petri dish not covered by the cells. The cantilever stiffness was determined to be 0.00664 N/m using the thermal tuning method in the air (*Hutter and Bechhoefer, 1993*). AFM indentation experiments were performed using cells in phosphate-buffered saline (PBS) solution in a 6 cm Petri dish maintained at a constant temperature of 37°C using a Petri dish heater. For imaging, a sufficient amount of PBS was used to maintain cell viability, and a 100 µl drop of PBS was placed on the AFM cantilever tip to avoid the formation of air bubbles between the cantilever holder and the sample. Force–displacement curves were acquired for each cell by indenting them at the central region of the cell, close to the nucleus. The force spectroscopy experiments were performed for a single force cycle by setting trigger points on the cantilever deflection (u) and tip velocity. A time gap of >45 s was incorporated between the indentation experiments within the same cell to account for stress relaxation. The force–displacement curve corresponding to the approach of the cantilever tip toward the substrate for each location was considered to quantify the effective cell modulus using the material properties of the cantilever and Hertzian contact mechanics model for a spherical indenting a flat plane (*Hertz, 1881*; *Guo and Akhremitchev, 2006*) According to this model, the force measured during cell indentation (F) is related to the cantilever indentation, δ, as (*Johnson, 1985*)

$$F = \frac{4ER_{tip}^{1/2}}{3\left(1-\mu^2\right)}\delta^{\frac{3}{2}}$$

where µ is the Poisson's ratio, E is the effective cell modulus, and $R_{tip}$ is the radius of the spherical cantilever tip. The cell is assumed to be incompressible with µ = 0.5. The force curves were delineated into a region prior to the contact point of the cantilever tip with the cell and the region after contact. Force data post-contact with the cell was used to calculate the effective cell modulus.

## FLIM imaging

Cells were placed in the incubator at 37°C in a humidified atmosphere containing 5% $CO_2$ for 20 min before imaging. For hyper-osmotic shock, MEF cells were treated with an 87 mM sucrose working solution in complete DMEM culture media devoid of phenol red and imaged after 15 min. Frequency-domain fluorescence lifetime imaging microscopy (FLIM) measurements were performed using a Nikon TE2000 confocal microscope with a 60×/1.2NA water immersion objective equipped with an Alba FastFLIM system (*Sun et al., 2011*). Specifically, cells stained with Flipper-TR (Cytoskeleton, CY-SC020) were excited using a 488 nm pulsed laser with a modulation frequency of 20 MHz and imaged through a 595/40 nm bandpass filter followed by MPD APD detectors. After image collection, bi-exponential fitting of FLIM images was performed using VistaVision software (ISS) to obtain fluorescent lifetimes ($\tau_1$ and $\tau_2$) of each pixel. Only the longest lifetime component ($\tau_1$) was used to represent relative membrane tension as described previously (*Colom et al., 2018*).

## Statistical analysis

Statistical analysis was performed with Prism 6 (GraphPad). An unpaired two-tailed Student's *t*-test or one-way ANOVA was used to determine the statistical significance of differences between groups. A p-value<0.05 was considered statistically significant.

Individual p-values are indicated in the figures, with no data points excluded from statistical analysis. One representative experiment of at leastthree independent repeats is shown.

## Acknowledgements

The work was supported by the National Institutes of Health (R35GM119787 to QD), (R35GM119785 to FH), (R01GM132501 to DU), and (P30CA023168 to Purdue Center for Cancer Research) for shared resources. This work is based upon efforts supported by EMBRIO Institute, contract #2120200, a National Science Foundation (NSF) Biology Integration Institute (to DC). YW is supported by Bisland Fellowship, Purdue University.

## Additional information

### Funding

| Funder | Grant reference number | Author |
| --- | --- | --- |
| National Institute of General Medical Sciences | R35GM119787 | Qing Deng |
| National Institute of Mental Health | R35GM119785 | Fang Huang |
| National Institute of General Medical Sciences | R01GM132501 | David Umulis |
| National Cancer Institute | P30CA023168 | Qing Deng |
| National Science Foundation | 2120200 | Deva Chan |
| Purdue University | Bisland Fellowship | Yueyang Wang |

The funders had no role in study design, data collection and interpretation, or the decision to submit the work for publication.

### Author contributions

Yueyang Wang, Conceptualization, Data curation, Formal analysis, Investigation, Visualization, Methodology, Writing – original draft, Writing – review and editing; Lee D Troughton, Data curation, Investigation, Methodology, Writing – original draft; Fan Xu, Investigation, Visualization, Methodology, Writing – original draft; Aritra Chatterjee, Han Zhao, Jingjuan Chen, Data curation, Investigation, Visualization, Methodology, Writing – original draft; Chang Ding, Tianqi Wang, Shelly Tan, Investigation; Laura Pulido Cifuentes, Investigation, Methodology, Writing – original draft; Ryan B Wagner, Linlin Li,

Investigation, Methodology; David Umulis, Data curation, Supervision; Shihuan Kuang, Deva Chan, Fang Huang, Supervision, Methodology; Daniel M Suter, Patrick W Oakes, Supervision, Methodology, Writing – original draft; Chongli Yuan, Supervision, Investigation, Methodology; Qing Deng, Conceptualization, Supervision, Funding acquisition, Writing – original draft, Writing – review and editing

## Author ORCIDs
Yueyang Wang http://orcid.org/0000-0002-5445-8186
Fan Xu https://orcid.org/0000-0001-6298-5587
Aritra Chatterjee https://orcid.org/0000-0002-5318-3459
Ryan B Wagner https://orcid.org/0000-0002-4111-8027
Linlin Li https://orcid.org/0000-0002-9667-2965
David Umulis https://orcid.org/0000-0003-1913-2284
Shihuan Kuang https://orcid.org/0000-0001-9180-3180
Daniel M Suter https://orcid.org/0000-0002-5230-7229
Chongli Yuan https://orcid.org/0000-0003-3765-0931
Deva Chan https://orcid.org/0000-0003-1508-1045
Patrick W Oakes https://orcid.org/0000-0001-9951-1318
Qing Deng https://orcid.org/0000-0002-9254-9951

## Decision letter and Author response
Decision letter https://doi.org/10.7554/eLife.88828.sa1
Author response https://doi.org/10.7554/eLife.88828.sa2

---

# Additional files

## Supplementary files
MDAR checklist

## Data availability
All data generated or analyzed during this study are included in the manuscript and supporting file; Source Data files have been provided for Figures 1-5, 7 and Figure 5-figure Supplement 1.

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

# Appendix 1

## Appendix 1—key resources table

| Reagent type (species) or resource | Designation | Source or reference | Identifiers | Additional information |
|---|---|---|---|---|
| Cell line (*Mus musculus*) | Wild-type MEF | ATCC | CRL-2991 | |
| Cell line (*M. musculus*) | *Mfn1*-null MEF | ATCC | CRL-2992 | |
| Cell line (*M. musculus*) | *Mfn2*-null MEF | ATCC | CRL-2993 | |
| Cell line (human) | HEK 293T/17 | ATCC | CRL-11268 | |
| Transfected construct (human) | MSCV-puro-Mfn1 | This paper | | MFN1 overexpression |
| Transfected construct (human) | MSCV-puro-Mfn2 | This paper | | MFN2 rescue |
| Transfected construct (human) | MSCV-puro-mito-GFP-ER | This paper | | Expressing the artificial tethering structure in MEFs |
| Transfected construct (human) | SPLICS Mt-ER Long P2A | Addgene | #164107 | |
| Transfected construct (*M. musculus*) | MSCV-puro-CaMKII-WT | This paper | | CaMKIIalpha expression |
| Transfected construct (*M. musculus*) | MSCV-puro-CaMKII-DN | This paper | | Dominant negative CaMKIIalpha expression |
| Transfected construct (*M. musculus*) | MSCV-puro-MLCK-CA | This paper | | Constitutive active MLCK expression |
| Transfected construct (*M. musculus*) | MSCV-puro-MRLC-GFP | This paper | | MRLC-GFP expression |
| Transfected construct | pCMV-dR8.2 dvpr | Addgene | #8455 | |
| Transfected construct | pCMV-VSV-G | Addgene | #8454 | |
| Transfected construct (human) | Lipofectamine 3000 | Invitrogen | L3000015 | Transfection reagent |
| Recombinant DNA reagent | PLKO.1-Puro-ctrl (SHC003) (plasmid) | Sigma-Aldrich | SHC 003 | Control plasmid for knock-down MEF lines |
| Recombinant DNA reagent | PLKO.1-Puro-shROCK (plasmid) | Sigma-Aldrich | TRCN0000022903 | Knockdown ROCK in MEFs |
| Recombinant DNA reagent | PLKO.1-Puro-shMLCK (plasmid) | Sigma-Aldrich | TRCN0000024037 | Knockdown MLCK in MEFs |
| Sequence-based reagent | MSCV-mfn2 insert F | This paper | PCR primers | CACGATAATACCATGGG CCACCATGTCCCTGCTC |
| Sequence-based reagent | MSCV-mfn2 insert R | This paper | PCR primers | TCTAGAGTCGCGGCCGCTTAC TTGTACAGCTCGTCCATGCC |
| Sequence-based reagent | MSCV-mfn1 insert R | This paper | PCR primers | TCGACTCTAGAGTCGCGGCCGCT TACTTGTACAGCTCGTCCATGCC |
| Sequence-based reagent | Mfn2 into plix-Nsil-F | This paper | PCR primers | AAAACCCCGGTCCTATGCATAT GTCCCTGCTCTTCTCTCGA |
| Sequence-based reagent | Mfn2 into plix-BamHI-R | This paper | PCR primers | CCCCAACCCCGGATCCTT ATCTGCTGGGCTGCAGGT |
| Sequence-based reagent | Camk2a-MSCV-F | This paper | PCR primers | AATTAGATCTCTCGAGGC CACCATGGTGAGCAAGG |
| Sequence-based reagent | Camk2a-MSCV-R | This paper | PCR primers | CTACCCGGTAGAATTCAT TCGGCGAAGCAAGAGCG |
| Sequence-based reagent | ER-mito F | This paper | PCR primers | AATTAGATCTCTCGAGATG GCAATCCAGTTGCGTTCG |
| Sequence-based reagent | ER-mito R | This paper | PCR primers | ATTTACGTAGCGGCCGCTTA AGATACATTGATGAGTTTGG |
| Sequence-based reagent | MRLC-GFP F | This paper | PCR primers | AATTAGATCTCTCGAGGC CACCATGGTGAGCAAGG |
| Sequence-based reagent | MRLC-GFP R | This paper | PCR primers | CTACCCGGTAGAATTCGCC CGCGGTCAGTCATCTTTG |
| Sequence-based reagent | MLCK-CA F | This paper | PCR primers | attagatctctcgagactagtcgactggatcc |
| Sequence-based reagent | MLCK-CA R | This paper | PCR primers | ccggtagaattcagatcttgggtgggttaattaa |
| Chemical compound, drug | BAPTA | Cayman Chemical | #11706 | |
| Chemical compound, drug | Y27632 | Cayman Chemical | #10005583 | |

*Appendix 1 Continued on next page*

*Appendix 1 Continued*

| Reagent type (species) or resource | Designation | Source or reference | Identifiers | Additional information |
|---|---|---|---|---|
| Chemical compound, drug | CK666 | Cayman Chemical | #29038 | |
| Chemical compound, drug | STO-609 acetate | Biotechne | #1551 | |
| Chemical compound, drug | A23187 | Cayman Chemical | #11016 | |
| Chemical compound, drug | Blebbistatin | Cayman Chemical | #13013 | |
| Chemical compound, drug | Oligomycin | Sigma-Aldrich | #495455 | |
| Chemical compound, drug | FCCP | Sigma-Aldrich | C2920 | |
| Chemical compound, drug | Rotenone | Sigma-Aldrich | #557368 | |
| Chemical compound, drug | Antimycin A | Sigma-Aldrich | A8674 | |
| Chemical compound, drug | CAS 1090893 | Millipore | #553511 | |
| Chemical compound, drug | RhoA inhibitor-I | Cytoskeleton, Inc | #CT-04 | |
| Chemical compound, drug | FK-506 | Cayman Chemical | #10007965 | |
| Chemical compound, drug | FAK14 | Cayman Chemical | #14485 | |
| Chemical compound, drug | BMS-5 | Cayman Chemical | #21072 | |
| Chemical compound, drug | ML-7 | Cayman Chemical | #11801 | |
| Chemical compound, drug | DAPI | Invitrogen | D1306 | 1 µg/ml for IF staining |
| Antibody | Anti-Mfn2 (rabbit polyclonal) | Cell Signaling Technology | 9482S | IF(1:200), WB (1:1000) |
| Antibody | Anti-Mfn1 (rabbit polyclonal) | Abcam | ab126575 | WB (1:1000) |
| Antibody | Anti-pan-CaMKII (rabbit polyclonal) | Cell Signaling Technology | #3362 | WB (1:1000) |
| Antibody | Anti-phosphor-CaMKII (Thr286) (rabbit polyclonal) | Cell Signaling Technology | #12716 | WB (1:1000) |
| Antibody | Anti-phosphor-myosin light chain 2 (Ser19) (rabbit polyclonal) | Cell Signaling Technology | #3671 | WB (1:1000) |
| Antibody | Anti-myosin light chain 2 (rabbit polyclonal) | Cell Signaling Technology | #3672 | IF (1:200), WB (1:1000) |
| Antibody | Anti-phospho-PAK (rabbit polyclonal) | Cell Signaling Technology | #2605S | WB (1:1000) |
| Antibody | Anti-PAK1/2/3 (rabbit polyclonal) | Cell Signaling Technology | #2604 | WB (1:1000) |
| Antibody | Anti-Vinculin (mouse monoclonal) | Sigma-Aldrich | #V9131 | WB (1:1000) |
| Antibody | HRP AffiniPure anti-rabbit IgG (goat polyclonal) | Jackson ImmunoResearch | #111-035-003 | WB (1:2500) |
| Antibody | Anti-mouse IgG Alexa Fluor 680 (goat polyclonal) | Invitrogen | #A28183 | WB (1:2500) |
| Antibody | Anti-rabbit IgG Alexa Fluor Plus 800 (goat polyclonal) | Invitrogen | #A32735 | WB (1:2500) |
| Antibody | Anti-rabbit Alexa Fluor 488, (chicken polyclonal) | Invitrogen | #A-21441 | IF (1:500) |
| Antibody | Anti-mouse Alexa Fluor 568, (Goat polyclonal) | Invitrogen | #A-11004 | IF (1:500) |
| Recombinant DNA reagent | PLKO.1-Puro (plasmid) | Sigma-Aldrich | RRID:Addgene_10878 | |
| Peptide, recombinant protein | Platelet-Derived Growth Factor-BB human | Sigma-Aldrich | P3201 | |
| Peptide, recombinant protein | AIP | R&D Systems | #5959/1 | |
| Sequence-based reagent | siRNA: nontargeting control | Thermo Fisher | 4390843 | Silencer Select |
| Commercial assay or kit | Rac1 Pull-Down Activation Assay Biochem Kit (Bead Pull-Down Format) | Cytoskeleton, Inc | #BK035 | |
| Commercial assay or kit | RhoA Pull-Down Activation Assay Biochem Kit (Bead Pull-Down Format) | Cytoskeleton, Inc | #BK036 | |
| Commercial assay or kit | Fluo-4 Calcium Imaging Kit | Invitrogen | F10489 | |
| Commercial assay or kit | Seahorse XF Cell Mito Stress Test Kit | Agilent Technologies | #103015-100 | |
| Commercial assay or kit | Flipper-TR | Cytoskeleton | CY-SC020 | |
| Commercial assay or kit | In-Fusion HD Cloning | Clontech | 639647 | |

*Appendix 1 Continued on next page*

*Appendix 1 Continued*

| Reagent type (species) or resource | Designation | Source or reference | Identifiers | Additional information |
|---|---|---|---|---|
| Software, algorithm | Python | This study | | See "Immunostaning and confocal imaging" and "traction force microscopy and analysis" |
| Software, algorithm | ImageJ software | ImageJ (http://imagej.nih.gov/ij/) | | |
| Software, algorithm | GraphPad Prism 6 | GraphPad Prism (https://graphpad.com) | | |

