## [Editor Report]

This important article presents evidence for a special role of mitofusin-2, not shared with mitofusin-1, in regulating the actin cytoskeleton through calcium, RhoA, and actomyosin contractility. The evidence is compelling and uses a variety of techniques, including creative approaches, to investigate mitofusin-2’s role in ER-mitochondrial tethering. The article will be of interest to investigators studying mitofusins, mitochondrial fission/fusion, and the actin cytoskeleton.

---

## [Decision Letter]

[Editors' note: this paper was reviewed by Review Commons.]

---

## [Author Response]

General Statements

First, we would like to reiterate the reviewer’s comment regarding the significance of our work:

R1: “There is a growing field of mitochondrial biology and how it relates to cell migration. This paper examines the function of a key mitochondrial morphology regulator, MFN2, and dissects the role for MFN2 in migration at the level of cytoskeletal regulation. We think that this is interesting and that it's clear that MFN2 has multiple functions in the cell.”

R2: “Delineating the role of Mfn2 in cell migration will represent a good, fundamental contribution to the field of cell migration. The reviewer finds this manuscript to be conceptually interesting.”

R3: “The main novelty here appears to be the connection between excess cytoplasmic calcium, MFN2 loss, and RhoA/myosin activation. This is interesting and a useful addition to the literature.”

We agree with the reviewers that the scope of our study is limited to a fundamental cell biology question of how mfn2 regulates the actin cytoskeleton. In particular, the formation of an atypical peripheral actin band. We have restructured the manuscript to focus better on cell biology. Our results do not directly address MFN2-related disease or the formation of peripheral actin bands in vivo. Nevertheless, our work contributes to a fundamental understanding of cell biology that sets the foundation for future work in MFN2-related disease mechanisms in vivo.

Point-by-point description of the revisionsReviewer #1:– Key conclusions that were less convincing:– RhoA and NMII are in the title as mechanistic downstream regulators of CaM, but the results in Figure 8 call into question the role of RhoA. Why does RhoA activation not influence cell size and circularity? Can you overexpress MRLC-GFP and inhibit Rho and restore the wt phenotype? The role of NMII is also not clear – why does overexpressing MLCK-CA not have a phenotype but overexpressing MLCK downstream target MRLC show the phenotype? Are there any alternative pathways to regulate MRLC? It's not being discussed or described in 6D's schematic.

Rho activation usually leads to the formation of more stress fibers and therefore does not lead to decreased cell size and increased circularity observed in MFN2 KO. The phenotype is restored by either ROCK or MLCK knockdown. We have discussed in the main text that the formation of PAB requires both RhoA and NMII activation under restricted spatiotemporal control.

MRLC has three major regulators (Ikebe and Hartshorne, 1985; Isotani et al., 2004). As we discussed, MLCK and ROCK phosphorylate MRLC at either Ser19 or Ser19 and Thr18. MRLC is dephosphorylated and inactivated by its phosphatase MLCP. We tried to knock down MLCP in wt MEF cells but failed to see any cell morphology changes (data not shown).

We were also surprised to see MRLC-GFP overexpression with Rho Activator can phenocopy PAB, but “MLCK-CA + Rho Activator” failed to. We believe it is because MLCK-CA constitutively over-activates a broad range of downstream effectors while overexpressing MRLC mimics endogenous activation or NMII alone. Also, only a proportion of cells acquired PAB structure under Rho Activator and MRLC overexpression, which indicates PAB formation also requires specific spatiotemporal controls.

Rewrite for clarity– The role of ER/mito contacts in the system was unclear (since ER/mito contacts were not observed nor evaluated directly).

We have included additional data to measure ER/mito contacts in MEFs. Our result is consistent with numerous previous reports that MFN2 regulates ER/mito contacts. The data is now included in Figure S3.

– What role does focal adhesions have on PAB formation or any part of the model – There were results showing larger focal adhesions in the MFN2-/- cells, but not sure how this fits in with the bigger picture of contractility and PABs, and focal adhesions were not in the model in Figure 5.

Focal adhesion and actomyosin are tightly coupled, and our work focuses on the actin network. Our model did not include FAs since FA is not a significant focus in this study.

– Whether regulating calcium impacts PAB formation

Calcium likely regulates PAB formation. We have shown PAB cell percentage decreases in mfn2-/- with ER-mito tethering contrast in Figure S3.

– The role of PABs in migration is also unclear – can you affect PAB formation or get rid of PABs and quantify cell migration?

Our data suggest that PAB formation and cell migration are inversely correlated. Since PAB results from a contractile actin band on the cell periphery, its role in defective cell spreading and migration is expected. We demonstrated that MLCK and ROCK knockdown reduced PABs and rescued cell spreading.

– It was hard to understand the correlation between the membrane tension of MFN2-/- cells and their ability to spread on softer substrates. How does this result fit in with the overarching model?

Reduced membrane tension is presumably associated with decreased cell spreading. Softer substrates attenuate the mechanical force on focal adhesion proteins and the actin cytoskeleton (Burridge and Chrzanowska-Wodnicka, 2003; Pelham and Wang, 1997; Wong et al., 2015), which is required for focal adhesion maturation. As a result, softer substrates can reduce the over-contraction in the MFN2 KO cells. The results support the model that MFN2 KO cells have enhanced cell contraction on the substrates dependent on substrate interaction and force transduction on focal adhesions.

Should the authors qualify some of their claims as preliminary or speculative, or remove them altogether?

We have removed the MFN2-related disease from the introduction to focus the paper more on cell biology in vitro.

– There were a number of findings that did not seem to fit in with the paper, or they were included, but were not robustly described nor quantified. As an example, Paxillin-positive focal adhesions were evaluated in MFN2-/- and with various pharmacological approaches, but there is no quantification with respect to size, number, or distribution of focal adhesions, despite language in the main text that there are differences between conditions.

We have quantified the focal adhesions in the KO cells, and the data is now included in Figure 5. We used the actin distribution to quantify the “PAB”; therefore, FAs are not a significant focus of this study.

Also, the model was presented in figure 5, and then there were several pieces of data presented afterward, some that are included in the model (myosin regulation), and some that are not in the model (membrane tension, TFM, substrate stiffness, etc).

Membrane tension and substrate stiffness dependence are physical properties of the cell. The model focuses on the molecular mechanism that leads to PAB formation.

The stiffness/tension figure was not clear to me, and it was difficult to make sense of the data since one would predict that an increase in actomyosin contractility at the cortex would lead to higher membrane tension, not lower, and then how membrane tension relates to spreading on soft matrices is also unclear.

The result was surprising to us initially. However, the MFN2 KO cells have increased actomyosin contractility only at the cell-substrate interface but not throughout the entire cell cortex. A less spread cell would have a more relaxed membrane and display a lower membrane tension, consistent with our observation. Softer matrices reduce cell contractility at the cell-substrate interface, which allows MFN2 KO cells to relax and spread better. We have emphasized in the discussion of our manuscript that MFN2 KO cells have an increase in actomyosin contractility only at the cell-substrate interface.

The manuscript seems like an amalgamation of different pieces of data that do not necessarily fit together into a cohesive story, so the authors are encouraged to either remove these data, or shore them up and weave them into the narrative.

We respectfully disagree with the reviewer since the cell morphology, actin structure, substrate interaction, and cell mechanics are tightly correlative and provide a complete picture of the role of MFN2 in regulating cell behavior.

Request additional experiments1. The imaging used for a lot of the quantification (migration, circularity) is difficult to resolve. The cells often look like they are not imaged in the correct imaging plane. It would be helpful to have better representative images such that it is clear how the cells were tracked and how cell periphery regions of interest were manually drawn. Focal adhesions should also be shown without thresholding.

We used a TIRF microscope and imaged a Z stack. Imaris was used to combine the Z-stack images. The images in the manuscripts are now of the lowest stack with background subtraction.

2. For most of the quantification, it appears that the experiment was only performed once and that a handful of cells were quantified. The figure legend indicates the number of cells (often reported as >12 cells or >25 cells), but the methods indicate that high content imaging was performed, and so the interpretation is that these experiments were only performed once. Biological replicates are required. If the data do represent at least 3 biological replicates, then more cell quantification is required (12 or 25 cells in total would mean quantifying a small number of cells per replicate).

We quantified more cells and indicated the number of cells quantified in the figure legends. The experiments are with three biological replicates.

3. Mitochondrial morphology and quantification should be performed in the MFN2 knockdown and rescue lines.

Mitochondrial morphology is well characterized in the Mfn2 KO and rescue MEF cells (Chen et al., 2003; Naon et al., 2016; Samanas et al., 2020). We observed a similar phenotype using mito-RFP to label mitochondrial structure (Figure S1).

4.Many of the comparisons throughout the figures is between MFN2 knockdown and MFN2 knockdown plus rescue or genetic/pharmacological approaches, but a comparison that is rarely made is between wildtype and experimental. These comparisons could be useful in comparing partial rescues and potential redundancies with the other mitofusin.

We have included the WT in our assays (Figure 2-6). We also confirmed that MFN1 could not rescue the MFN2 defects (Figure 2). We observed partial and complete rescue in different assays. It would be difficult to conclude whether the phenotype is due to the redundancies with the other mitofusin because not all cells are rescued at the endogenous level.

5. For the mito/ER tethering experiments, it is important to show that ER/mito contacts are formed and not formed in the various conditions with imaging approaches.

We adopted a previously established method to quantify ER-mitochondria contacts with the probe SPLICSL (Cieri et al., 2017; Vallese et al., 2020). Our results align with previous reports that Mfn2-null MEFs displayed significantly decreased ER-mitochondria contacts. MFN2 re-expression or ER-mitochondria tethering structure restored the contacts. (Figure S3).

6. For some of the pharmacological perturbations, it would be helpful to show that the perturbation actually led to the expected phenotype – as an example, in cells treated with different concentrations of A23187, what are the intracellular calcium levels and how do these treatments influence PAB formation? This aspect should be generally applied across the study – when a modification is made, that particular phenotype should first be evaluated, before dissecting how the perturbation affects downstream phenotypes.

We selected a collection of well-characterized inhibitors broadly used in the literature for pharmacological perturbations. For example, numerous studies used A23187 treatment to raise intracellular calcium to examine related actin cytoskeleton changes (Carson et al., 1994; Goldfine et al., 1981; Shao et al., 2015). We titrated the drugs in WT in preliminary experiments and observed similar phenotypes. (data not shown). We then use the same concentration to treat the MFN2 KO cells. Overall, we use pharmacological perturbations as supporting evidence. We use genetics (knockdown or overexpression) to validate our results.

7. In Figures 4 and 5, the thresholding approaches in the images make the focal adhesions difficult to resolve, and therefore it is difficult to determine the size. As described above, these metrics should be defined and quantified.

We used a TIRF microscope and imaged a Z stack. Imaris was used to combine the Z-stack images. The images in the manuscripts are now of the lowest stack with background subtraction.

8. What is a PAB? How is it defined? What metrics make a structure a PAB versus regular cortical actin – are there quantifiable metrics? In figure 8, there are some structures that are labelled as a PAB, but some aren't (as an example, the left panel in 8b is a PAB, but the right panel in 8A is not, but they look the same), so a PAB should be defined with quantifiable measures, and then applied to the entire study.

We developed an algorism to quantify PAB cells. We first used the ImageJ plugin FiloQuant (Jacquemet et al., 2019) to identify the cell border and cytoskeleton, then used our custom algorism to quantify the percentage of actin in the cell border area. The cellular circularity is also calculated at the same time. If the cell contains more than 50% actin in the cell border area, and the circularity is higher than 0.6, we then count it as a “PAB” cell (Figure S2).

9. As described above, why does RhoA activation not influence cell size and circularity? Can you overexpress MRLC-GFP and inhibit Rho and restore the wildtype phenotype?

Rho activation usually leads to the formation of more stress fibers and therefore does not lead to decreased cell size and increased circularity observed in MFN2 KO.

We are sorry that we don’t understand the rationale of this experiment proposed by the reviewer. ROCK inhibition restored the wildtype phenotype in MFN2 KO cells (Figure 7). Figure 8 is to create the MFN2 KO phenotype in WT cells, which requires both Rho and MRLC overactivation.

10. Are the data and the methods presented in such a way that they can be reproduced?-We appreciate that the authors quantified many parameters, although some quantifications were missing. There are some missing methods – how was directionality quantified, was migration quantified by selecting the approximate center of cells using MTrackJ or were centroids quantified by outlining cells, for instance. Also, given that some of the phenotypes were somewhat arbitrarily assigned (ie. what constitutes a PAB?), it may be difficult to reproduce these approaches and interpret data appropriately.

We have clarified directionality quantification methods and other details. We used MTrackJ to track cell migration. And as we mentioned above, we came up with a customized algorithm to quantify PAB cells, which shows the critical effectors in a more quantifiable way.

11. Are the experiments adequately replicated and statistical analysis adequate?-Unfortunately, while the approximate number of cells was reported, the number of biological replicates were not reported, and therefore, the experimental information and statistical analyses are not adequate.

We have quantified more cells and indicated the number of repeats and cells quantified in the figure legend.

Minor comments:Specific experimental issues that are easily addressable.12. For some of the graphs – mostly about calcium levels – fold change is shown, but raw values should also be included to determine whether the basal levels of calcium are different across the conditions.

Δ F/F_0_ is the standard method to normalize dye loading in cells for calcium

concentration measurements (Kijlstra et al., 2015; Zhou et al., 2021).

13. Scale bars in every panel should also help make the points clearer.

We have added scale bars in all panels.

Reviewer #2 (Evidence, reproducibility and clarity (Required)):1. Figure 1A: The Mfn1 Western Blot is not of publication quality. Moreover, quantitation is necessary.

We performed additional western blots, changed the representative images, and quantified the level of knockdown and overexpression (Figure 2 and 7). We did not quantify the WB in Figure 1A since it was to confirm that the Mfn1-/- or Mfn2-/- were knock-out cell lines.

2. Figure 1B (as well as Figure 2G and others): the date do not reflect cellular size but instead spread cellular area.

We thank the reviewer for this suggestion. We have changed all similar descriptions to “Spread Area” in the main text and figures.

3. Figure 1C, D: Mfn1-null MEFs appear to be more spindle-shaped than wt cells, yet their circularity tends to be elevated. Do the authors have an explanation?

The circularity of Mfn1-/- MEFs has a slight increase but is not significant compared to the wt cells. As we observed, Mfn1-/- MEFs have fewer protrusions than wt, which may contribute to the slight increase in its circularity (Figure 5C). However, this is not the focus of this study.

4. Figure 2A: The Mfn1 levels in Mfn2-/- + Mfn2 are lower than Mfn2-/-? Does this imply a crosstalk between Mfn1 and Mfn2 expression.

We agree with the reviewer that a compensatory change in MFN1 expression might happen in Mfn2-/- + MFN2 MEFs. Previous research also indicated crosstalk between MFN1 and MFN2 expression (Sidarala et al., 2022).

5. Figure 2H: The authors should provide co-staining of mitochondria and Mfn2 as well.

Co-staining of mitochondria and MFN2 in Mfn2-/- MEFs or rescue lines has been done in numerous previous studies (Chen et al., 2003; Naon et al., 2016; Samanas et al., 2020). In this work, we transfected our cells with mito-RFP and showed mitochondria changes in Mfn2-/- and rescue MEF cells (Figure S1G).

6. Figure 4D-E: Western blots are not of publication quality. Looking at the blots provided in Figure 4D, the reviewer is not convinced with the quantitative data shown in Figure 4E. For instance, the intensity of pCaMKII band for "vec" does not look 3x higher than that of "+MFN2", whereas that of "+MFN2" looks much higher than that of WT.

We have performed additional western blots and changed the representative images.

7. Figure 5C: The authors should stain for vinculin, which are present in mature FAs only, rather than paxillin which are present in all FAs. This would strengthen the authors' conclusions. Also, FA size should be quantified.

We have quantified FA size in Figure 5. The maturity of FAs is not a major focus of this study. It is likely that most FAs here are mature since they are connected with stress fibers.

8. Figure 6C – Why does the background have a grid and appear grey in color? Also, the cell interior appears in different colors in the different images. The authors should take a z-stack of images and provide the raw image files.

We used a TIRF microscope and imaged a Z stack. Imaris was used to combine the Z-stack images. The images in the manuscripts are now of the lowest stack with background subtraction.

9. Figure 7C: The MLCII Western blot is not of publication quality, and may affect the quantification provided in Figure 7D.

We have performed additional western blots and changed the representative images.

10. Figure 8: Do cell treatments with Rho Activator and MLCK-CA also impair migration velocity similar to Mfn2-null cells?

Our data indicated that Rho Activator and MRLC induced the “PAB” structure seen in MFN2 KO cells. It is likely that cell migration is impaired here. Spatiotemporal regulation of Rho Activation is important to cell migration, it is known that Rho overactivation can significantly inhibit cell migration (Nobes and Hall, 1999). Showing Rho Activator and MLCK-CA will reduce cell migration will not add new knowledge to the cell migration field. However not all cell migration defects are associated with the PAB. We, therefore, focused on PAB quantification in this figure.

11. Figure 9A: The authors claim that wt cells have actin bundles that protrude against the membrane while Mfn2-null cells do not. This does not look convincing as the Mfn2-null actin bundle seem to be pushing against the membrane at the bottom of the image. No quantification is provided. It is unclear what conclusion can be drawn from the super-resolution images.

We used super-resolution imaging to demonstrate the details of the peripheral actin band (PAB) structure. We have used two boxes to enlarge the regions where membrane parallel actin structures are predominant. The quantification of PAB is provided in other figures.

12. Suppl. Figure 5C: The authors should take images using a confocal microscope for cells with Flipper-TR construct, eliminate the background and the cell center to only consider the cell periphery. Nikon TE2000 does not seem to be a confocal microscope.

The amount of Flipper-TR that MEF cells can take in was limited. With the current signal-to-noise ratio, complete background elimination is not feasible. A confocal microscope is not necessary for Flipper-TR FLIM imaging (García-Calvo et al., 2022).

Reviewer #3 (Evidence, reproducibility and clarity (Required)):General Comments (Major)1. Data Presentation and analysis:The data analysis would benefit from using a method such as Super-Plots to show the data from separate biological repeats and to use N numbers that represent the number of biological repeats rather than the number of cells analysed. Please see the following reference for a suggestion on how to analyse the data:Lord SJ, Velle KB, Mullins RD, Fritz-Laylin LK. SuperPlots: Communicatingreproducibility and variability in cell biology. J Cell Biol. 2020 Jun1;219(6):e202001064. doi: 10.1083/jcb.202001064. PMID: 32346721; PMCID:PMC7265319.

We have changed the dot colors to show data from separate biological repeats.

2. Another general comment is that many of the experiments show analysis of very few numbers of cells or maybe only one field of view in a microscopy quantification experiment. This seems unusually low – for example, in Figure 1E only 6 cells have been analysed. It seems like more could have been done and if statistical analysis like we suggest in 1 above is used, this might reveal that some of the differences are less significant than the authors think/report. This is important, as cells are noisy and it is unusual to have such high significance for experiments like cell migration and other parameters unless a lot of measurements are made. In Figure 1G, it appears that only one field of view has been used to quantify the data. We routinely use 3-5 fields of view to get a representative sample of what the cells are doing.

We have quantified more cells and indicated the number of repeats and cells quantified in the figure legend.

3. Some of the micrographs appear to be missing scale bars- e.g. Figure 2H, Figure 8.

We have included scale bars in the lower right corner of all panels.

4. In the cell tracking experiments, only some of the cells in the images appear to have been tracked. How were the tracked cells chosen? Normally, we would track every cell to avoid bias in selection.

We tracked all the cells in the view at the beginning of the experiments.

5. The western blot images do not show the molecular size of the bands. Show ladder position.

We have added bands to show molecular weights.

6. Mostly the graphs show individual data points, which is good, but in some cases only a bar is shown- it would be nice to have individual points overlaid on the bars- e.g. Figure 1I, 4E, 5B, 5E, 7B, 7D

We have updated the graph to show individual points.

7. Many of the confocal images look very processed- they have no background and have a hazy black halo around the cell. I am not familiar with the type of processing that was done and I worry that the images are only showing a masked and processed version of the actual data. The authors need to explain what processing they have done and probably also to provide the unprocessed images in a supplementary figure or dataset for readers to see. The methods description is inadequate as it only says Image J was used to process the data.

We used a TIRF microscope and imaged a Z stack. Imaris was used to combine the Z stack images. The images in the manuscripts are now of the lowest stack with background subtraction.

Individual comments on Figures:Figure 1:See general comments above- consider to use Superplots, more cells and more fields of view in quantifications. Show experimental points in bar graph.

We have quantified more cells and used super plots to display the data. The number of repeats and cells quantified are indicated in the figure legend.

Figure 2:In 2E, the colours are very similar for two of the experiments so it is difficult to distinguish them- e.g. the MFN1 vs MFN2 rescue data both appear dark blue.

We have changed the color for MFN1 rescue to distinguish the two samples better.

In 2H are the magnifications really the same for the WT as the +DOX and -DOX? The cell in the WT looks huge. Is this representative? Also, the phalloidin stain looks very spotty on the WT. This seems unusual.

The images are of the same scale. The Mfn2-/- MEFs are smaller, and DOX-induced MFN2 expression can only partially rescue the cell size.

Figure 3:Not many cells were analysed in 3B, especially the zero time point.

We have quantified more cells.

Please define +T, we assume it is the tether construct, but it is not defined

We defined T as a tether in the main text and the figure legend.

In 3F, how were the tracked cells chosen?

We tracked all the cells in the view at the beginning of the experiments.

Figure 4:4B: Why have they not tested FK506 and STO609 on the WT cells?

We focused on understanding the MFN2 KO phenotype. Since neither FK506 nor STO609 altered the MFN2 KO phenotype, we did not include them in the WT group.

4C: How were the tracked cells chosen?

We tracked all the cells in the view at the start of the experiments.

4D-E: The blot doesn't look representative of the quantification- were the numbers normalised to vinculin? The difference between WT and vector looks too large to be real if the amounts were normalised to the vinculin, as vinculin is increased in vector. Likewise, the pCAMKII looks to be substantially decreased from the +MFN2, but this is not what the quantification shows.

We have performed additional western blots and changed the representative images.

Figure 5:5B- please clarify which ratio is shown here. I assume it is the ratio of RhoA-GTP vs RhoA between the zero and 4 minute time points.

Yes, we have clarified this point in the figure legend.

5C- These images appear to have a mask around the cell. It is hard to tell where the edge of the cell really is- what sort of processing was used? Especially for the paxillin staining, why is there no cytoplasm shown? Is this because the image is in TIRF?

We used a TIRF microscope and imaged a Z stack. Imaris was used to combine the Z-stack images. The images in the manuscripts are now of the lowest stack with background subtraction.

Figure 6:Figure 6C- the blebbistatin treated cell looks very large- is this representative?

Yes, Blebbistatin-treated cells are larger (Figure 6A).

Figure 7:Figure 7C- The lanes for MLCII are all run together- is this from a different gel? Is this quantification accurate?

We have performed additional western blots and changed the representative images.

Figure 7F- What is the % level of knockdown achieved?

The level of knockdown is labeled on the figure.

Figure 8:Figure 8A,B- does the scale bar represent all of the images in these two panels?

Yes, the figure legend is updated to clarify this point.

Figure 8C,D- Superplots would be helpful here.

We have used super plots to display the data.

Supplementary Data:The OCR data do not add much and are not discussed much in the manuscript. Perhaps they could be omitted.

Our OCR data ruled out the possibility of metabolic regulation. Since MFN2 is a mitochondria protein with its typical functions in metabolic pathways, we cannot omit its influence on metabolism here. As we observed, shMLCK enhanced OCR, shROCK reduced OCR, and both knock-down rescued cell morphology and motility. We believe that PAB formation is independent of MFN2’s function in metabolic regulation.

Figure S3- The figure label doesn't match the manuscript test- was fibrinogen or collagen used?

We tried cover glass alone, collagen, and fibronectin-coated glass. The PAB formation is independent of these extracellular substrates. We did not try fibrinogen because MEF cell is reported to prefer fibronectin (Lehtimäki et al., 2021).

Reference

Burridge, K., and Chrzanowska-Wodnicka, M. (2003). FOCAL ADHESIONS, CONTRACTILITY, AND SIGNALING. Http://Dx.Doi.Org/10.1146/Annurev.Cellbio.12.1.463, 12, 463–519. https://doi.org/10.1146/ANNUREV.CELLBIO.12.1.463

Carson, S. D., Perry, G. A., and Pirruccello, S. J. (1994). Fibroblast Tissue Factor: Calcium and Ionophore Induce Shape Changes, Release of Membrane Vesicles, and Redistribution of Tissue Factor Antigen in Addition to Increased Procoagulant Activity. Blood, 84(2), 526–534. https://doi.org/10.1182/BLOOD.V84.2.526.526

Chen, H., Detmer, S. A., Ewald, A. J., Griffin, E. E., Fraser, S. E., and Chan, D. C. (2003). Mitofusins Mfn1 and Mfn2 coordinately regulate mitochondrial fusion and are essential for embryonic development. The Journal of Cell Biology, 160(2), 189. https://doi.org/10.1083/JCB.200211046

Cieri, D., Vicario, M., Giacomello, M., Vallese, F., Filadi, R., Wagner, T., Pozzan, T., Pizzo, P., Scorrano, L., Brini, M., and Calì, T. (2017). SPLICS: a split green fluorescent protein-based contact site sensor for narrow and wide heterotypic organelle juxtaposition. Cell Death and Differentiation 2018 25:6, 25(6), 1131–1145. https://doi.org/10.1038/s41418-017-0033-z

García-Calvo, J., López-Andarias, J., Maillard, J., Mercier, V., Roffay, C., Roux, A., Fürstenberg, A., Sakai, N., and Matile, S. (2022). HydroFlipper membrane tension probes: imaging membrane hydration and mechanical compression simultaneously in living cells. Chemical Science, 13(7), 2086–2093. https://doi.org/10.1039/D1SC05208J

Goldfine, S. M., Schroter, E. H., and Izzard, C. S. (1981). Calcium-dependent shortening of fibroblasts induced by the ionophore, A23187. Journal of Cell Science, 50(1), 391–405. https://doi.org/10.1242/JCS.50.1.391

Ikebe, M., and Hartshorne, D. J. (1985). Phosphorylation of Smooth Muscle Myosin at Two Distinct Sites by Myosin Light Chain Kinase*. Journal of Biological Chemistry, 260, 10027–10031. https://doi.org/10.1016/S0021-9258(17)39206-2

Isotani, E., Zhi, G., Lau, K. S., Huang, J., Mizuno, Y., Persechini, A., Geguchadze, R., Kamm, K. E., and Stull, J. T. (2004). Real-time evaluation of myosin light chain kinase activation in smooth muscle tissues from a transgenic calmodulin-biosensor mouse. Proceedings of the National Academy of Sciences of the United States of America, 101(16), 6279–6284. https://doi.org/10.1073/PNAS.0308742101

Kijlstra, J. D., Hu, D., Mittal, N., Kausel, E., van der Meer, P., Garakani, A., and Domian, I. J. (2015). Integrated Analysis of Contractile Kinetics, Force Generation, and Electrical Activity in Single Human Stem Cell-Derived Cardiomyocytes. Stem Cell Reports, 5(6), 1226. https://doi.org/10.1016/J.STEMCR.2015.10.017

Lehtimäki, J. I., Rajakylä, E. K., Tojkander, S., and Lappalainen, P. (2021). Generation of stress fibers through myosin-driven reorganization of the actin cortex. *ELife*, 10, 1–43. https://doi.org/10.7554/*ELIFE*.60710

Naon, D., Zaninello, M., Giacomello, M., Varanita, T., Grespi, F., Lakshminaranayan, S., Serafini, A., Semenzato, M., Herkenne, S., Hernández-Alvarez, M. I., Zorzano, A., De Stefani, D., Dorn, G. W., and Scorrano, L. (2016). Critical reappraisal confirms that Mitofusin 2 is an endoplasmic reticulum-mitochondria tether. Proceedings of the National Academy of Sciences of the United States of America, 113(40), 11249–11254. https://doi.org/10.1073/PNAS.1606786113/SUPPL_FILE/PNAS.201606786SI.PDF

Nobes, C. D., and Hall, A. (1999). Rho GTPases Control Polarity, Protrusion, and Adhesion during Cell Movement. The Journal of Cell Biology, 144(6), 1235. https://doi.org/10.1083/JCB.144.6.1235

Pelham, R. J., and Wang, Y. L. (1997). Cell locomotion and focal adhesions are regulated by substrate flexibility. Proceedings of the National Academy of Sciences of the United States of America, 94(25), 13661. https://doi.org/10.1073/PNAS.94.25.13661

Samanas, N. B., Engelhart, E. A., and Hoppins, S. (2020). Defective nucleotide-dependent assembly and membrane fusion in Mfn2 CMT2A variants improved by Bax. Life Science Alliance, 3(5). https://doi.org/10.26508/LSA.201900527

Shao, X., Li, Q., Mogilner, A., Bershadsky, A. D., and Shivashankar, G. V. (2015). Mechanical stimulation induces formin-dependent assembly of a perinuclear actin rim. Proceedings of the National Academy of Sciences of the United States of America, 112(20), E2595–E2601. https://doi.org/10.1073/PNAS.1504837112/SUPPL_FILE/PNAS.1504837112.SM03.AVI

Sidarala, V., Zhu, J., Levi-D’Ancona, E., Pearson, G. L., Reck, E. C., Walker, E. M., Kaufman, B. A., and Soleimanpour, S. A. (2022). Mitofusin 1 and 2 regulation of mitochondrial DNA content is a critical determinant of glucose homeostasis. Nature Communications 2022 13:1, 13(1), 1–16. https://doi.org/10.1038/s41467-022-29945-7

Vallese, F., Catoni, C., Cieri, D., Barazzuol, L., Ramirez, O., Calore, V., Bonora, M., Giamogante, F., Pinton, P., Brini, M., and Calì, T. (2020). An expanded palette of improved SPLICS reporters detects multiple organelle contacts in vitro and in vivo. Nature Communications, 11(1). https://doi.org/10.1038/S41467-020-19892-6

Wong, S. Y., Ulrich, T. A., Deleyrolle, L. P., MacKay, J. L., Lin, J. M. G., Martuscello, R. T., Jundi, M. A., Reynolds, B. A., and Kumar, S. (2015). Constitutive activation of myosin-dependent contractility sensitizes glioma tumor-initiating cells to mechanical inputs and reduces tissue invasion. Cancer Research, 75(6), 1113–1122. https://doi.org/10.1158/0008-5472.CAN-13-3426

Zhou, W., Hsu, A. Y., Wang, Y., Syahirah, R., Wang, T., Jeffries, J., Wang, X., Mohammad, H., Seleem, M. N., Umulis, D., and Deng, Q. (2021). Mitofusin 2 regulates neutrophil adhesive migration and the actin cytoskeleton. Journal of Cell Science, 133(17). https://doi.org/10.1242/JCS.248880/VIDEO-11